# BOOSTING DATASET DISTILLATION WITH THE ASSISTANCE OF CRUCIAL SAMPLES

## ABSTRACT

In recent years, massive datasets have significantly driven the advancement of machine learning at the expense of high computational costs and extensive storage requirements. Dataset distillation (DD) aims to address this challenge by learning a small synthetic dataset such that a model trained on it can achieve a comparable test performance as one trained on the original dataset. This task can be formulated as a bi-level learning problem where the outer loop optimizes the learned dataset and the inner loop updates the model parameters based on the distilled data. Different from previous studies that focus primarily on optimizing the inner loop in this bi-level problem, we delve into the task of dataset distillation from the perspective of sample cruciality. We find that discarding easy samples and keeping the hard ones that are difficult to be represented by the learned synthetic samples in the outer loop can be beneficial for DD. Motivated by this observation, we further develop an Infinite Semantic Augmentation (ISA) based dataset distillation algorithm, which discards some easier samples and implicitly enriches harder ones in the semantic space through continuously interpolating between two target feature vectors. Through detailed mathematical derivation, the joint contribution to training loss of all interpolated feature points is formed into an analytical closed-form solution of an integral that can be optimized with almost no extra computational cost. Experimental results on several benchmark datasets demonstrate the effectiveness of our approach in reducing the dataset size while preserving the accuracy of the model. Furthermore, we show that high-quality distilled data can also provide benefits to downstream applications, such as continual learning and membership inference defense. The code can be found at `https://github.com/to_be/released`.

## 1 INTRODUCTION

Deep learning has achieved remarkable success in various fields including computer vision (Rombach et al., 2022; He et al., 2022), neural language processing (OpenAI, 2023; Taori et al., 2023), protein structure prediction (Bordin et al., 2023; Lin et al., 2023) thanks to the recent advances in technology and availability of massive real-world data. However, the storage and usage of these data can be very resource-intensive and time-consuming (Lei & Tao, 2023). Therefore, the need for efficient and scalable methods for handling and processing large datasets has become increasingly pressing. Dataset distillation (DD) addresses this challenge by learning a small set of synthetic examples from a large dataset such that a model trained on it can achieve a comparable test performance as one trained on the original dataset (Zhou et al., 2022; Loo et al., 2023). In this way, not only can the storage and training budgets be reduced, but the highly condensed and synthetic nature can also benefit various downstream applications, such as continual learning (De Lange et al., 2021), neural architecture search (Elsken et al., 2019), and privacy-preserving tasks (Park et al., 2022).

Dataset distillation was firstly studied from the perspective of matching the characteristic of the original dataset, such as distribution matching (DM) (Zhao & Bilen, 2023; Zhao et al., 2023), gradient matching (GM) (Zhao et al., 2021), training trajectory matching (MTT) (Cazenavette et al., 2022; Du et al., 2022), *etc.*. For example, MTT (Cazenavette et al., 2022) proposes to match segments of parameter trajectories trained on synthetic data with segments of pre-recorded trajectories from models trained on original data and thus avoid being short-sighted or difficult to optimize. While these matching-based methods require pre-defined surrogate objectives that may introduce some bias and

may not accurately reflect the true objective (Zhou et al., 2022), the recently developed meta-learning based approaches (Loo et al., 2023; 2022; Nguyen et al., 2021) treat the dataset distillation as a bi-level optimization problem with an inner objective to update model parameters on the synthetic set and an outer (meta) objective to refine the distilled sets via meta-gradient updates. With the nested loop, the synthetic dataset gradually converges towards one of the optimal solutions. However, the inner loop for updating model parameters is typically achieved through multiple steps of gradient descent for neural networks, making the whole process quite time-consuming. To address this issue, many researchers have made considerable efforts. One representative work can be the kernel ridge regression (KRR) (Song et al., 2022) method, which replaces the neural network in the inner loop with a kernel model and bypasses the recursive back-propagation of the meta-gradient for efficiency.

In this paper, different from studies that focus on optimizing the strategy of the inner loop, we explore what kind of target data is crucial for DD in the outer loop. As the storing budget is quite limited (even only one synthetic sample per class), it is almost impossible to preserve all the information in the original dataset. This makes us wonder whether we should make a compromise to focus on learning from the majority of data points around the center to ensure the good performance on these data. On the contrary, what would happen if we discard some of these samples? To answer these questions, we conduct extensive experiments and find that samples that are difficult to be represented by the learned synthetic samples in the outer loop are more crucial for DD. Based on this observation, we further develop an infinite semantic augmentation (ISA) based dataset distillation algorithm, which discards some easier samples and implicitly enriches harder ones in the semantic space through continuously interpolating between two target feature vectors to reduce the influence of the majority of samples around the center. It is worth noting that the joint contribution to training loss of all interpolated feature points is formed into an analytical closed-form solution of an integral, which means we can utilize all interpolated semantic features for dataset distillation with almost no extra computational costs. Extensive experiments on several benchmark datasets show that the proposed method can enhance the dataset distillation performance as well as the applications in both continual learning and privacy-preserving tasks. To this end, we make three major contributions:

- We explore the dataset distillation from the perspective of crucial samples and experiments show that hard samples are more valuable for this task.
- Based on the above finding, we develop an infinite semantic augmentation based dataset distillation algorithm that considers an infinite number of virtual samples between two real samples without extra computational cost.
- We show the effectiveness of the proposed method in dataset distillation tasks, as well as applications, including continual learning and privacy preservation.

## 2 METHOD

### 2.1 DATASET DISTILLATION AS BI-LEVEL OPTIMIZATION

Given a large labeled dataset $\mathcal{T} = \{(\mathbf{x}_i, \mathbf{y}_i)\}_{i=1}^{|\mathcal{T}|}$ where $\mathbf{x}_i$ is the $i$-th image and $\mathbf{y}_i$ is the corresponding label. $|\mathcal{T}|$ is the number of samples in this dataset. Denote the expected risk of model $f$ parameterized by $\theta$ on data distribution $\mathcal{D}$ as $\mathcal{R}_{\mathcal{D}}(\theta)$:

$$\mathcal{R}_{\mathcal{D}}(\theta) = \mathbb{E}_{(\mathbf{x},\mathbf{y})\sim\mathcal{D}}[\mathcal{L}(f_\theta(\mathbf{x}), \mathbf{y})]. \tag{1}$$

where $\mathcal{L}(f_\theta(\mathbf{x}), \mathbf{y})$ is the objective function for computing validation loss. The goal of dataset distillation (DD) is to learn a small synthetic dataset $\mathcal{S} = \{(\mathbf{x}_i, \mathbf{y}_i)\}_{i=1}^{|\mathcal{S}|}$ ($|\mathcal{S}| \ll |\mathcal{T}|$) so that the test performance of models trained on $\mathcal{S}$ is similar to that on $\mathcal{T}$. That is, the expectation of model's validation loss $\mathcal{L}$ trained with algorithm $\mathcal{A}(\mathcal{S}, \theta^0)$ by the training set $\mathcal{S}$ under different initialized network parameter $\theta^0$ from model pool $P_\theta$ should be similar to that of $\mathcal{T}$:

$$\mathbb{E}_{\theta^0\sim P_\theta}[\mathcal{R}_{\mathcal{D}}(\mathcal{A}(\mathcal{T}, \theta^0))] \simeq \mathbb{E}_{\theta^0\sim P_\theta}[\mathcal{R}_{\mathcal{D}}(\mathcal{A}(\mathcal{S}, \theta^0))]. \tag{2}$$

Since the data distribution $\mathcal{D}$ is unknown, a practical way to estimate the expected risk is by the empirical risk $\mathcal{R}_{\mathcal{T}}(\theta)$. In consequence, the objective of DD can be converted to minimize $\mathcal{R}_{\mathcal{T}}(\mathcal{A}(\mathcal{S}))$. To this end, the dataset distillation can be formulated as the following bi-level optimization problem:

$$\underbrace{\mathcal{S}^* := \underset{\mathcal{S}}{\arg\min}\, \mathbb{E}_{\theta^0\sim P_\theta}[\mathcal{R}_{\mathcal{T}}(\mathcal{A}(\mathcal{S}, \theta^0))],}_{\text{outer loop}} \quad \text{where} \quad \underbrace{\mathcal{A}(\mathcal{S}, \theta^0) := \underset{\theta}{\arg\min}\, \mathcal{R}_{\mathcal{S}}(\theta^0).}_{\text{inner loop}} \tag{3}$$

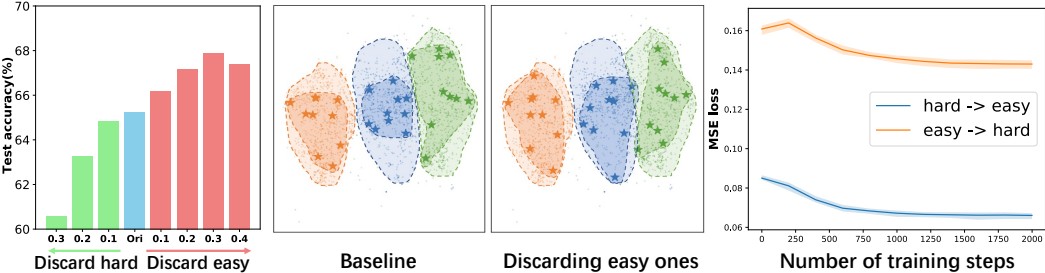

Figure 1: Left: Accuracy performances on training networks under different situations. "Ori" indicates the original results. We first discard the 10%, 20%, 30% samples with the largest MSE loss in each batch to drop the hardest samples (green). The performance gets dropped compared to the original ones (blue). In contrast, when the easiest samples are discarded (red), the performances get a boost. Middle: Distributions of synthetic images learned by the original baseline and baseline with discarding easy samples. The orange, blue, green points are the real images of three classes while the stars are the corresponding learned synthetic images. Right: "hard -> easy" indicates adopting the harder samples to make predictions of easier ones with Eq. 4 and vice versa.

The inner loop optimizes the model parameters based on the synthetic dataset $\mathcal{S}$. During the outer loop iteration, the synthetic set is optimized by minimizing the model's risk in terms of the target dataset $\mathcal{T}$. With the nested loop, the synthetic dataset gradually converges towards one of the optimal solutions. In the following paper, we denote $\mathcal{R}_{\mathcal{T}}(\mathcal{A}(\mathcal{S}, \theta^0)) = \mathcal{L}(\mathcal{A}(\mathcal{S}, \theta^0), \mathcal{T})$ for simplify and the meta-training loss is usually defined as:

$$\mathcal{L}(\mathcal{A}(\mathcal{S}, \theta^0), \mathcal{T}) = \frac{1}{2}||Y_{\mathcal{T}} - K^{\theta}_{X_{\mathcal{T}} X_{\mathcal{S}}}(K^{\theta}_{X_{\mathcal{S}} X_{\mathcal{S}}} + \lambda I)^{-1}Y_{\mathcal{S}}||_2^2, \quad s.t. \quad \theta = \mathcal{A}(\mathcal{S}, \theta^0). \quad (4)$$

$(X_{\mathcal{T}}, Y_{\mathcal{T}})$ and $(X_{\mathcal{S}}, Y_{\mathcal{S}})$ are the inputs and labels of target data $\mathcal{T}$ and distilled synthetic data $\mathcal{S}$ respectively. $\lambda$ controls the regularization strength. The conjugate kernel is defined by the inner product of the neural network features as $K^{\theta}_{X_{\mathcal{T}} X_{\mathcal{S}}} = f_{\theta}(X_{\mathcal{T}})f_{\theta}(X_{\mathcal{S}})^T$.

## 2.2 DATASET DISTILLATION WITH CRUCIAL SAMPLES

As illustrated in Eq. 3 that the inner loop optimizes the model parameters based on the synthetic dataset $\mathcal{S}$, it requires multiple steps of gradient descent for neural networks and is quite time-consuming. Therefore, previous studies focus primarily on optimizing the inner loop of the above bi-level problem (Song et al., 2022; Loo et al., 2022; Zhou et al., 2022; Loo et al., 2023). Different from these researches, we delve into the task of DD from the perspective of sample cruciality in the outer loop and pay attention to exploring what kind of samples from the original dataset are critical for DD. Since the number of distilled samples is quite limited (even only one per class), it is almost impossible to preserve all the information in the original dataset. This makes us wonder whether we should make a compromise to focus on the majority of data points around the center to ensure good performances on these data. On the contrary, what would happen if we discard these samples?

To answer these questions, we conduct experiments on CIFAR10 dataset (Krizhevsky et al., 2009) (IPC=10 where IPC is the number of images per class) by discarding $p * 100\%$ percent target samples ($p \in (0, 1)$) in terms of sample hardness according to their meta-training loss in Eq. 4. Higher loss indicates greater hardness. As shown in Figure 1 (Left), we first discard the 10%, 20%, 30% samples with the largest loss from Eq. 4 in each batch to drop the hardest samples ($p = 0.1, 0.2, 0.3$) that are difficult to be represented by the learned synthetic ones. It can be observed that the test accuracy in this scenario (green) gets a drop compared to the original ones (blue). On the contrary, when we drop the easiest 10% of samples, the model's performance (red) gets a boost on all settings. These indicate that the hard samples are more crucial for dataset distillation.

To investigate this phenomenon, we explore it from the perspective of data manifold and information. Regarding the data manifold, we show the distributions of synthetic images learned by the original baseline and baseline with discarding easy samples in Figure 1 (Middle). The results demonstrate an increased overlap between the generated images (represented by stars) and the original dataset (represented by dots) after discarding some easy samples, thereby depicting a better representation of the

---

**Algorithm 1** Dataset distillation with the Assistance of Crucial Samples

---

**Require:** $\mathcal{T}$: the target labeled dataset; $p$: discard percent; $\epsilon$: distance controlling parameter
1:  Initialization: $\mathcal{S} = (X_{\mathcal{S}}, Y_{\mathcal{S}})$, a model pool $\mathcal{M}$ with $m$ models $\{\theta_i\}_{i=1}^{m}$
2:  **while** no converged **do**
3:     Sample a model uniformly from the model pool $\mathcal{M}$: $\theta_i \sim \mathcal{M}$
4:     Sample a target batch uniformly from the labeled dataset: $(X_{\mathcal{T}}, Y_{\mathcal{T}}) \sim \mathcal{T}$
5:     Compute the meta-training loss $\mathcal{L}$ using Eq. 4
6:     Discard $p * 100\%$ percent easiest samples according to $\mathcal{L}$ to get the new $X_{\mathcal{T}}$
7:     Compute the new loss $\mathcal{L}_a$ with Eq. 7 as the joint contribution of infinite many virtual samples
8:     Update the distilled data $\mathcal{S}$ : $X_{\mathcal{S}} \leftarrow X_{\mathcal{S}} - \alpha \nabla \mathcal{L}_a$, and $Y_{\mathcal{S}} \leftarrow \alpha \nabla \mathcal{L}_a$
9:     Train the model $\theta_i$ on the current distilled data $\mathcal{S}$
10:    Reinitialize the model $\theta_i$ if $\theta_i$ has been updated more than $K$ steps
11: **end while**

---

manifold. As for the information perspective, to find out whether the harder samples have contained information in easier samples, we divide the target data into two splits and use the harder ones to make predictions for easier ones with Eq. 4. The MSE loss is 0.0661, as shown in Figure 1(Right). This loss indicates that the information in harder samples is enough for making precise predictions for easier samples. In contrast, when we use the easier samples to predict the harder ones, the loss is 0.1430. Therefore, harder samples cannot be replaced by simple samples. We also step further to compare the diversity of synthetic images with the recall value, which is a commonly used metric in generative tasks for evaluating the diversity of generative model (Sajjadi et al., 2018; Naeem et al., 2020). Higher recall indicates greater diversity. After incorporating the process of discarding easier samples in the outer loop, the recall value notably grows from 0.84 to 0.89. This improvement suggests an enhanced diversity in the synthetic samples. More details can be found in the supplementary.

## 2.3 CRUCIAL SAMPLE EXTENSION WITH SEMANTIC AUGMENTATION

The above finding reveals that reducing redundancy in easy samples and taking more crucial samples into consideration can be beneficial for improving the diversity of synthetic samples and better depicting the data manifold in the dataset distillation tasks. Therefore, adopting sample extension methods to create more valuable virtual samples may help for improving task performance. Actually, there have been various data extension methods such as CutMix (Yun et al., 2019), MixUp (Zhang et al., 2017) that have demonstrated to be effective in enhancing the performance of deep learning models across various tasks such as image classification, object detection, *etc.*. However, previous studies have found that data augmentations during the dataset distillation can induce instability (Zhou et al., 2022; Loo et al., 2023) and introduce little help. As a result, seldom do they adopt these strategies in the training procedure of this task. Many studies show that the application failures of MixUp can be caused by the low-quality problem in the input space and propose to conduct MixUp in feature layers to alleviate this issue (Verma et al., 2019; Venkataramanan et al., 2022). However, never have them been validated in the DD task. Furthermore, similar to the vallina MixUp, they still consider only one virtual sample per forward pass, leading to potential inefficiencies.

To address the aforementioned challenges, we put forward an infinite semantic augmentation method that enables the augmentation of an infinite number of virtual samples in the semantic space by continuously interpolating between two target feature vectors. Denote $\mathbf{z} = f_\theta(\mathbf{x})$ as the corresponding feature vectors of $\mathbf{x}$ extracted by $f$, which is parameterized with $\theta$. Let $\mathbf{z}'$ represent the feature vector of another sample $\mathbf{x}'$. Given a hyperparameter $\alpha \in [0, \epsilon]$ where $\epsilon \in [0, 1]$ controls the distance between virtual samples and the start point, we augment the training data in the feature space by linearly interpolating between $\hat{\mathbf{z}} = \alpha \mathbf{z} + (1 - \alpha)\mathbf{z}'$, and integrate the loss function with respect to $\alpha$. This approach allows us to consider a wide range of virtual samples simultaneously, which is different from usual MixUp methods that sample one $\alpha$ per batch. Furthermore, the contribution of augmented samples to the loss function can be calculated analytically, which does not require extra computational costs. In the following, we introduce the calculation of the loss function in detail.

Denote $\omega = (K_{X_{\mathcal{S}} X_{\mathcal{S}}}^{\theta} + \lambda I)^{-1} Y_{\mathcal{S}}$, then the loss for a single sample $(\mathbf{x}, \mathbf{y}) \sim \mathcal{T}$ in Eq. 4 is

$$\mathcal{L}_s(\mathbf{x}, \mathbf{y}) = \frac{1}{2}||\mathbf{y} - f_\theta(\mathbf{x}) f_\theta(X_{\mathcal{S}})^T \omega||_2^2. \tag{5}$$

Table 1: Distillation performance in term of test accuracy (%) on several benchmark datasets.

| | IPC | DSA | DM | KIP | RFAD | MTT | FRePo | Ours |
|---|---|---|---|---|---|---|---|---|
| | 1 | 88.7±0.6 | 89.9±0.8 | 90.1±0.1 | **94.4±1.5** | 91.4±0.9 | 93.0±0.4 | 93.2±0.4 |
| MNIST | 10 | 97.9±0.1 | 97.6±0.1 | 97.5±0.0 | 98.5±0.1 | 97.3±0.1 | **98.6±0.1** | 98.5±0.1 |
| | 50 | 99.2±0.1 | 98.6±0.1 | 98.3±0.1 | 98.8±0.1 | 98.5±0.1 | **99.2±0.0** | **99.2±0.0** |
| | 1 | 70.6±0.6 | 71.5±0.5 | 73.5±0.5 | **78.6±1.3** | 75.1±0.9 | 75.6±0.3 | 77.1±0.4 |
| F-MNIST | 10 | 84.8±0.3 | 83.6±0.2 | 86.8±0.1 | 87.0±0.5 | **87.2±0.3** | 86.2±0.2 | 86.0±0.2 |
| | 50 | 88.8±0.2 | 88.2±0.1 | 88.0±0.1 | 88.8±0.4 | 88.3±0.1 | **89.6±0.1** | 89.5±0.1 |
| | 1 | 36.7±0.8 | 31.0±0.6 | 49.9±0.2 | **53.6±1.2** | 46.3±0.8 | 46.8±0.7 | 48.4±0.4 |
| CIFAR-10 | 10 | 53.2±0.8 | 49.2±0.8 | 62.7±0.3 | 66.3±0.5 | 65.3±0.7 | 65.5±0.4 | **67.2±0.4** |
| | 50 | 66.8±0.4 | 63.7±0.5 | 68.6±0.2 | 71.1±0.4 | 71.6±0.2 | 71.7±0.2 | **73.8±0.0** |
| | 1 | 16.8±0.2 | 12.2±0.4 | 15.7±0.2 | 26.3±1.1 | 24.3±0.3 | 28.7±0.1 | **31.2±0.2** |
| CIFAR-100 | 10 | 32.3±0.3 | 49.2±0.8 | 28.3±0.1 | 33.0±0.3 | 40.1±0.4 | 42.5±0.2 | **46.4±0.5** |
| | 50 | 42.8±0.4 | 43.6±0.4 | - | - | 47.7±0.2 | 44.3±0.2 | **49.4±0.3** |
| T-ImageNet | 1 | 6.6±0.2 | 3.9±0.2 | - | - | 8.8±0.3 | 15.4±0.3 | **19.8±0.1** |
| | 10 | - | 12.9±0.4 | - | - | 23.2±0.2 | 25.4±0.2 | **27.0±0.3** |

The join contribution to training loss of all interpolated feature points between $(\mathbf{x}, \mathbf{y})$ and another data point $(\mathbf{x}', \mathbf{y}')$ is

$$\mathcal{L}_a(\mathbf{x}, \mathbf{y}) = \frac{1}{\epsilon} \int_0^\epsilon ||(\mathbf{z}+\alpha\delta_\mathbf{1})X_\mathcal{S}\omega - (\mathbf{y}+\alpha\delta_\mathbf{2})||_2^2 d\alpha = \frac{1}{\epsilon} \int_0^\epsilon ||(\mathbf{z}X_\mathcal{S}\omega - \mathbf{y}) + \alpha(\delta_\mathbf{1}X_\mathcal{S}\omega - \delta_\mathbf{2})||_2^2 d\alpha, \tag{6}$$

where $\delta_\mathbf{1} = \mathbf{z}' - \mathbf{z}$, $\delta_\mathbf{2} = \mathbf{y}' - \mathbf{y}$.

Denote $A = \mathbf{z}X_\mathcal{S}\omega - \mathbf{y}$ and $B = \delta_\mathbf{1}X_\mathcal{S}\omega - \delta_\mathbf{2}$, we have

$$\mathcal{L}_a(\mathbf{x}, \mathbf{y}) = A^2 + \epsilon AB + \frac{1}{3}\epsilon^2 B^2. \tag{7}$$

We recognize that the first item is the original outer-loop loss of sample $(\mathbf{x}, \mathbf{y})$. The last two terms summarize the contribution of all samples interpolating between $\mathbf{x}$ and $\mathbf{x}'$. It is straightforward to verify that the last two terms vanish when $\epsilon = 0$. The whole algorithm can be found in Algorithm 1.

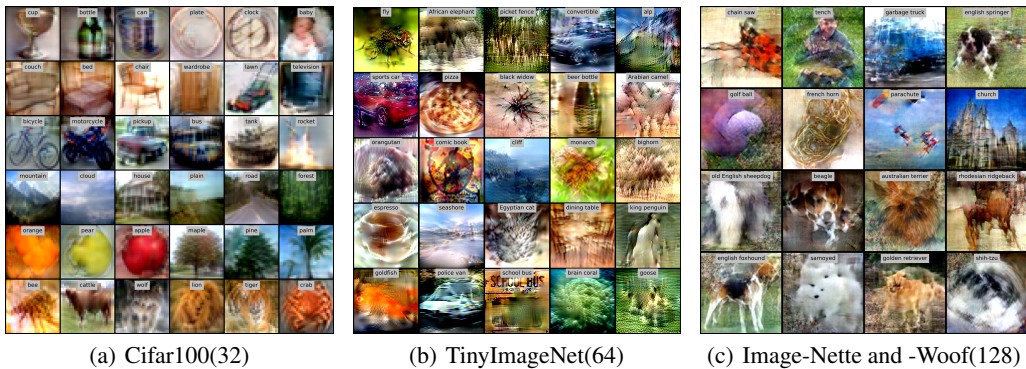

(a) Cifar100(32)  (b) TinyImageNet(64)  (c) Image-Nette and -Woof(128)

Figure 2: Example distilled images. 32, 64, 128 are the corresponding resolutions.

## 3 EXPERIMENTS

**Implementation Details.** We eastablish our algorithm based on FRePo (Zhou et al., 2022) by default and compare our method to several state-of-the-art dataset distillation methods on various benchmark datasets. $\epsilon = 1.0$ and $p = 0.2$ for all experiments. Following other methods (Nguyen et al., 2021; Zhou et al., 2022), the $\lambda$ is set to be $10^{-6}\text{Tr}(K^\theta_{X_\mathcal{S}X_\mathcal{S}})$. All the distilled data are evaluated using five random initialized neural networks and we report the mean and standard deviation.

Table 2: Test accuracy (%) performance of CIFAR-10 with 10 images per class distilled datasets evaluated on different network architectures. NN, DN, IN, and BN stand for no normalization, default normalization, Instance Normalization, Batch Normalization respectively. FRePo* refers to the results reproduced with the officially released code by distilling the datasets and evaluating on them.

| | Train Arch | Evaluation Architecture | | | | | |
| --- | --- | --- | --- | --- | --- | --- | --- |
| | | Conv | Conv-NN | ResNet-DN | ResNet-BN | VGG-BN | AlexNet |
| DSA | Conv-IN | 53.2±0.8 | 36.4±1.5 | 42.1±0.7 | 34.1±1.4 | 46.3±1.3 | 34.0±2.3 |
| DM | Conv-IN | 49.2±0.8 | 35.2±0.5 | 36.8±1.2 | 35.5±1.3 | 41.2±1.8 | 34.9±1.1 |
| MTT | Conv-IN | 64.4±0.9 | 41.6±1.3 | 49.2±1.1 | 42.9±1.5 | 46.6±2.0 | 34.2±2.6 |
| KIP | Conv-NTK | 62.7±0.3 | 58.2±0.4 | 49.0±1.2 | 45.8±1.4 | 30.1±1.5 | 57.2±0.4 |
| FRePo* | Conv-BN | 65.9±0.4 | 65.9±0.4 | 53.2±0.8 | 50.9±0.6 | **55.4±0.5** | 62.3±0.6 |
| Ours | Conv-BN | **67.2±0.4** | **67.2±0.4** | **54.5±1.3** | **51.1±1.3** | 54.6±0.5 | **64.9±0.2** |

Table 3: Distillation performance in term of test accuracy (%) for ImageNet subsets.

| IPC | ImageNette (128x128) | | ImageWoof (128x128) | | ImageNet (64x64) | |
| --- | --- | --- | --- | --- | --- | --- |
| | 1 | 10 | 1 | 10 | 1 | 2 |
| Random Subset | 23.5±4.8 | 47.7±2.4 | 14.2±0.9 | 27.0±1.9 | 1.1±0.1 | 1.4±0.1 |
| MTT (Cazenavette et al., 2022) | 47.7±0.9 | 63.0±1.3 | 28.6±0.8 | 35.8±1.8 | - | - |
| FRePo (Zhou et al., 2022) | 48.1±0.7 | 66.5±0.8 | 29.7±0.6 | 42.2±0.9 | 7.5±0.3 | 9.7±0.2 |
| Ours | **49.6±0.6** | **67.8±0.3** | **30.8±0.5** | **43.8±0.6** | **8.0±0.2** | **10.7±0.1** |

## 3.1 STANDARD BENCHMARKS

**Distillation Performance.** We first evaluate our method on five standard benchmark datasets including MNIST (10 classes) (LeCun et al., 1998), Fashion-MNIST (10 classes) (Xiao et al., 2017), CIFAR10 (10 classes) (Krizhevsky et al., 2009), CIFAR100 (100 classes) (Krizhevsky et al., 2009), Tiny-ImageNet (200 classes) (Le & Yang, 2015). Following the same setting with most previous methods, the number of synthetic images per class (IPC) is set to be 1, 10, 50 for the first four datasets and 1, 10 for Tiny-ImageNet. We compare our method with six baseline dataset distillation algorithms including both matching-based methods like Differentiable Simese Augmentation (DSA) (Zhao & Bilen, 2021), Distribution Matching (DM) (Zhao & Bilen, 2023), Matching Training Trajectories (MTT) (Cazenavette et al., 2022) and bi-level optimization based method including Kernel-Inducing-Points (KIP) (Nguyen et al., 2021), Random Feature Approximation (RFAD) (Loo et al., 2022), neural Feature Regression (FRePo) (Zhou et al., 2022). The results are listed in Table 1. It can be observed that our method can achieve the state-of-the-art performance, validating the effectiveness of the proposed method. The synthetic images can be found in Figure 2.

**Cross-architecture Generalization.** One desirable characteristic of distilled datasets is the ability to generalize effectively to unseen training architectures. Therefore, we evaluate the generalization ability of our distilled datasets on CIFAR10 under 10 images per class setting. Following prior work, we evaluate our models on the ResNet-18 (He et al., 2016), VGG11 (Simonyan & Zisserman, 2014), and AlexNet (Krizhevsky et al., 2017) with evaluation on various normalization layers such as using no normalization (NN), batch normalization (BN) (Ioffe & Szegedy, 2015), and instance normalization (IN) (Ulyanov et al., 2016). As shown in Table 2, our approach can achieve high generalization ability while we hold a good performance on the original architecture.

**Experiments on ImageNet Dataset.** With the rapid development in computer vision, a good performance on higher-resolution datasets is crucial for practice. As such, we step further to explore the performance on ImageNet (Deng et al., 2009). In line with FRePo (Zhou et al., 2022) and MTT (Cazenavette et al., 2022), we consider two ImageNet subsets: ImageNette and Image-Woof (Howard). Both of them consist of 10 classes with a resolution of 128x128. The results in Table 3 demonstrate that our approach is capable of enhancing the performance of our baseline FRePo. To evaluate how well the proposed method scales to more complex label spaces, we also consider the full ImageNet-1K dataset with 1000 classes (Deng et al., 2009), resized to 64x64. The results in Table 3 also indicate the practicability of our proposed methodology.

## 3.2 ABLATION STUDIES.

**Ablation Studies on Each Proposed Module.** To validate the effectiveness of the proposed Infinite Semantic Augmentation (ISA) and the discarding strategy, we add each component individually to the baseline method. Results in Figure 3(a) show that both components can help to improve the performances. Furthermore, we show the performances under various $\epsilon$ to verify the effectiveness of the proposed ISA for virtual sample extension. As shown in Figure 3(b), the test accuracy gets a boost from 65.5% to 66.4% as the $\epsilon$ increases. This makes sense since larger step sizes can take more virtual samples into consideration. We also compare the performance with the MixUp strategy, as shown in Figure 3(c), the proposed extension method (66.4%) can hold a better performance than the vallina MixUp (65.81%). When compared with the single step-based semantic augmentation whose test accuracy is 65.95%, our approach still holds an advantage. These indicate the effectiveness of the proposed extension method. Besides, we observe that the proposed ISA can also enhance the generalization ability of synthetic images, which is witnessed by Figure 3(d) as expected since ISA can be regarded as a regularizer that can alleviate the overfitting problem.

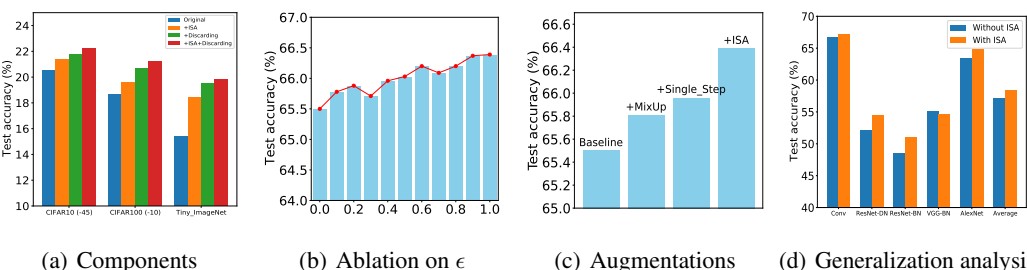

|      (a) Components      |      (b) Ablation on $\epsilon$      |      (c) Augmentations      |      (d) Generalization analysis      |

Figure 3: Ablation studies of individual components.

**Effectiveness on Other Methods** In this paper, we provide a concise and effective approach to improve the performance of the existing dataset distillation methods. We mainly show the effectiveness of applying our module on FRePo. Since the proposed method focuses on exploring what kind of target data from the original dataset is more crucial for dataset distillation. It is orthogonal to methods that developed with different bi-level optimization algorithms. Therefore, we also evaluate the performances of combining our module with various state-of-the-art dataset distillation methods such as RFAD (Loo et al., 2022), RCIG (Loo et al., 2023) in Table 4, which suggests that the proposed method can bring improvements and indicates the practicability of our method. More experiments on matching-based methods can be found in Appendix (Section F).

Table 4: Test accuracies of applying our module to other DD methods. * indicates the results are reproduced with officially released codes.

| IPC | Method | RFAD* | FRePo* | RCIG* |
|---|---|---|---|---|
| 1 | Normal | 52.1±0.1 | 46.8±0.7 | 53.9±0.5 |
|   | +Ours | **54.8±0.0** | **48.4±0.4** | **54.2±0.3** |
| 10 | Normal | 65.3±0.1 | 65.5±0.4 | 67.3±0.3 |
|    | +Ours | **67.0±0.1** | **67.2±0.4** | **68.0±0.6** |
| 50 | Normal | 69.8±0.2 | 71.7±0.2 | 73.5±0.2 |
|    | +Ours | **70.2±0.2** | **73.8±0.0** | **73.8±0.4** |

## 3.3 APPLICATIONS

**Continual Learning:** Given the high condensed nature of synthetic data generated by dataset distillation, they could serve as a critical component of continual learning algorithm (De Lange et al., 2021) that can help to mitigate the issue of catastrophic forgetting problem by condensing the past knowledge into a replay buffer. Indeed, there have been successful applications of dataset distillation in the continual learning scenario (Zhao & Bilen, 2021; 2023). To find out whether the proposed method can do some help in this scenario, we follow MTT (Zhao & Bilen, 2021) and FRePo (Zhou et al., 2022) that set up the baseline based on GDumb (Prabhu et al., 2020) which greedily stores class-balanced training examples in memory and train model from scratch on the latest memory only. In this case, the continual learning performance is solely reliant on the quality of the replay buffer. In our experiment, we conduct class-incremental learning on CIFAR100 with 5 and 10 step increments, while gradually increasing the buffer size by 20 images per class. To ensure consistency, we follow the same class division as MTT and FRePo and compare our method with approaches including

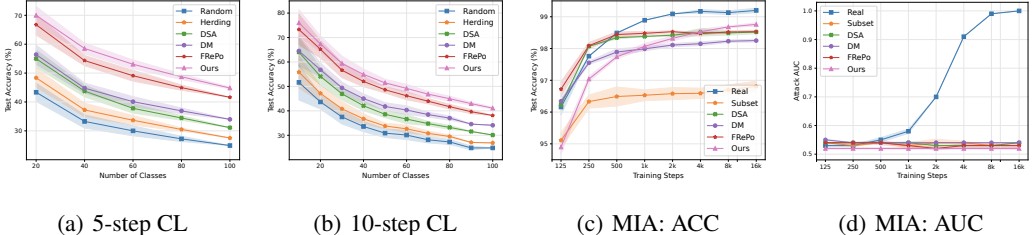

|         | (a) 5-step CL | (b) 10-step CL | (c) MIA: ACC | (d) MIA: AUC |
|---------|---------------|----------------|--------------|--------------|

Figure 4: (a,b) Multi-class accuracies (↑) across all classes observed up to a certain time point in Continual Learning (CL). (c,d) Test accuracy (↑) and attack AUC (↓) during each training steps. As the AUC is high in real data trained scenario, it is relatively small in distilled data trained scenarios.

Table 5: AUC of five attackers on models trained on the real and distilled data. The model trained on the real data is vulnerable to MIAs, while the model trained on the distilled data is robust to MIAs.

|        | Test Acc(%)      | Attack AUC | | | | |
|--------|------------------|-----------------|-----------------|-----------------|-----------------|-----------------|
|        |                  | Threshold       | LR              | MLP             | RF              | KNN             |
| Real   | **99.2 ± 0.1**   | 0.99 ± 0.01     | 0.99 ± 0.00     | 1.00 ± 0.00     | 1.00 ± 0.00     | 0.97 ± 0.00     |
| Subset | 96.8 ± 0.2       | 0.52 ± 0.00     | **0.50 ± 0.01** | 0.53 ± 0.01     | 0.55 ± 0.00     | 0.54 ± 0.00     |
| DSA    | 98.5 ± 0.1       | **0.50 ± 0.00** | 0.51 ± 0.00     | 0.54 ± 0.00     | 0.54 ± 0.01     | 0.54 ± 0.01     |
| DM     | 98.3 ± 0.0       | **0.50 ± 0.00** | 0.51 ± 0.01     | 0.54 ± 0.01     | 0.54 ± 0.01     | 0.53 ± 0.01     |
| FRePo  | 98.5 ± 0.1       | 0.52 ± 0.00     | 0.51 ± 0.00     | 0.53 ± 0.01     | **0.52 ± 0.01** | **0.51 ± 0.01** |
| Ours   | 98.8 ± 0.0       | 0.52 ± 0.00     | **0.50 ± 0.01** | **0.52 ± 0.00** | 0.53 ± 0.00     | 0.53 ± 0.00     |

random sampling (Prabhu et al., 2020), herding (Chen et al., 2012), DSA (Zhao & Bilen, 2021), DM (Zhao & Bilen, 2023), and FRePo (Zhou et al., 2022).

Figure 4(a) 4(b) shows that our method outperforms all previous methods. The final accuracy of all classes for our method and the second best method (FRePo) are 44.83% and 41.61% in 5-step learning while they are 41.10% and 38.09% for 10-step learning. This indicates that a better dataset distillation method can benefit for improving the application performance in continual learning.

**Privacy Preservation:** Membership inference attack (MIA) is a kind of privacy-related attack that involves an attacker attempting to determine whether a specific data record has been used in training the target model (Hu et al., 2022). It is particularly concerning in situations where sensitive data is used to train the model, such as in healthcare or financial applications. Therefore, we hope a model is trained to learn from the training data instead of memorizing them to preserve the privacy of the training data. However, previous studies have shown that deep neural networks are vulnerable to membership inference attacks and may leak the privacy of their training set (Shokri et al., 2017). Given that dataset distillation is designed to compress the dataset while retaining model performance, the generated synthetic images can be highly-condensed and may help to preserve the privacy. Therefore, in this section, we will explore whether the model trained on the distilled dataset will help to preserve the privacy from leaking. We repeat the experimental procedure of FRePo, distill 10000 images of MNIST to 500 images and conduct five popular "black box" membership inference attacks provided by Tensorflow Privacy (tf) to the model trained on these 500 distilled images. The attack methods include a threshold attack and four model-based attacks using logistic regression (LR), multi-layer perceptron (MLP), random forest (RF) and K-nearest neighbor (KNN). These attacks take the ground-truth labels, model prediction and losses as inputs and output whether the given data has been used for training. Apart from the test accuracy during training, we also adopt the commonly used metric AUC (the area under the ROC curve) of the attack classifier as the evaluation metrics to measure the privacy vulnerability of the trained model. Higher test accuracy, lower AUC indicate better performance. Following prior work (Zhao & Bilen, 2023; 2021), we keep a balanced set of training examples (member) and test examples (non-member) with 10K each to maximize the uncertainty of MIA. Thus, the random guessing strategy results in a 50% MIA accuracy.

As shown in Figure 4(c) 4(d) and Table 5, while models trained on the distilled data can hold a comparable test accuracy with real data trained ones, their attack AUCs are close to random guessing.

In contrast, model trained on real data are easy to be attacked. These indicate that dataset distillation can help to preserve the privacy of training data. We observe that among all the competitors, our approach can improve both the test accuracy and privacy-preserving performances. An interesting phenomenon is that while most competitors can get a test accuracy boost from the early stage, the models trained on real data and our distilled data seem to grow slowly and require more training steps to reach the best performance. We hypothesize that this is caused by the information contained in training data are really rich, thus requiring more steps for training.

## 4 RELATED WORK

**Coresets and Dataset distillation.** Coresets (Pooladzandi et al., 2022; Yang et al., 2023) are weighted subsets of the training data such that training on them results in the similar performance to training on the full dataset. In contrast, instead of selecting subsets of the training data, dataset distillation generates synthetic samples. It can be grouped into data matching framework (Zhao & Bilen, 2023; Cazenavette et al., 2022; Shin et al., 2023; Wang et al., 2022; Kim et al., 2022) and meta-learning framework based on the objectives applied to mimic target data. For example, distribution matching (DM) (Zhao & Bilen, 2023) is one of the representative of the former one, which optimizes the distilled data by aligning the distribution of synthetic data with that of target data. Zhao *etc.* (Zhao & Bilen, 2022) further improves the DM by changing the optimization from the input space to latent space with a well-trained generative adversarial network (GAN) to produce the synthetic examples. In contrast, the recently developed meta-learning based approaches treat the dataset distillation as a bi-level optimization problem with an inner objective to update model parameters on the synthetic set and an outer (meta) objective to refine the distilled sets via meta-gradient updates. Representative works include KIP (Nguyen et al., 2021), RFAD (Loo et al., 2022), FRePo (Zhou et al., 2022), *etc.*. The main difference between these bi-level methods mainly lies in the optimization methods used in the inner loop. In this paper, we focus on the outer loop where we study what kind of target data is more crucial for dataset distillation to boost the performance of dataset distillation.

**Data Extension.** There have been various methods for data extension during training, such as CutOut (DeVries & Taylor, 2017), CutMix (Yun et al., 2019), *etc.*. Among them, MixUp (Zhang et al., 2017) can be one of the representative. By generating new training examples via linearly combining pairs of existing examples in the input space and their corresponding labels, MixUp has demonstrated its efficacy in enhancing the generalization ability of deep learning models. Although being effective in most tasks such as image classification, object detection, input mixup images are overlays and tend to be unnatural, limiting its performance in dataset distillation tasks (Loo et al., 2023; Zhou et al., 2022). Recently, Bengio *et al.* (Verma et al., 2019) show that traversing along the manifold of representations obtained from feature layers can result in finding realistic examples. However, same with the vallina MixUp, it only considers one virtual sample for each forward pass, making it less effective. In contrast, we put forward an infinite semantic augmentation method that can take all the virtual samples between the two real samples with only one forward pass, requiring no extra computational costs while being effective.

## 5 DISCUSSIONS, LIMITATIONS, AND CONCLUSION

In this paper, we propose to investigate what kind of data are crucial for dataset distillation and experimental results show that it is the hard sample that matters more. Based on this observation, we further propose an infinite semantic augmentation method to create more virtual crucial samples for better performance, which augments an infinite number of samples in the semantic space by continuously interpolating between two target feature vectors. The joint contribution to the training loss of all interpolated feature points is formed into an analytical closed-form solution of an integral that can be optimized with almost no extra computational costs. Experimental results show that our method can not only improve the performances in dataset distillation but can also benefit downstream applications including continual learning and membership inference defense.

This paper reveals that target data selection matters for the dataset distillation. In the future work, it will be interesting to investigate some pre-processing methods that can transform the target dataset into a middle-size one by selecting or creating a few representative prototype examples from the original training dataset, such as coreset selection, condensing the target dataset gradually, *etc.*.

ETHICS STATEMENT

In this paper, we propose to investigate what kind of data is crucial for dataset distillation. Based on our observations, we further introduce an infinite semantic augmentation method to boost performance of dataset distillation. We hope this work can provide some insights to the community and help to reduce the cost on storage in today's era of big data. We did not use crowdsourcing and did not conduct research with human subjects in our experiments. We cited the creators when using existing assets (*e.g.*, code, data, models).

REPRODUCIBILITY STATEMENT

Our proposed module is an appealingly simple method, which is easy to be adopted in publicly available codes. We specify the settings of hyper-parameters and how they were chosen in our paper. The source code for our method can be found in our supplementary materials.

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

## A  APPENDIX

## B  DATASET DISTILLATION WITH CRUCIAL SAMPLES

In this section, we add more experiments showing the influence of discarding easy or hard examples on other datasets. The results are shown in Figure 5. There is a performance boost when easier samples are discarded at a small rate (red) while dropping the hardest ones can hurt the performances (green). Recall that the goal for dataset distillation is to condense the large dataset into a smaller one such that a model trained on it can achieve a comparable test performance as one trained on original dataset. Namely, we need to depict a good decision boundary for classification based on the distilled data. However, relying on easy samples solely may result in short-cut learning Geirhos et al. (2020). In contrast, hard samples are hard-to-be-distinguished samples that usually exist along the decision boundary thus can help to support a more explicit decision boundary. We think that's why hard samples can help to improve the dataset distillation performances.

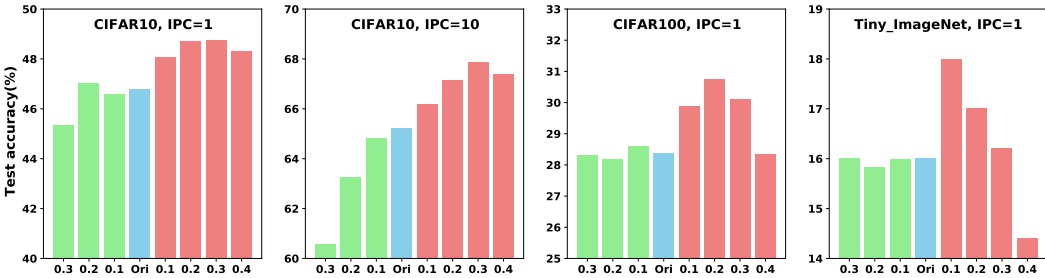

Figure 5: Accuracy performances on training networks under different situations. "Ori" indicates the original results. We first discard the 10%, 20%, 30% samples with the largest MSE loss in each batch to drop the hardest samples (green). The performance gets dropped compared to the original ones (blue). In contrast, when the easiest samples are discarded (red), the performances get a boost.

To investigate this phenomenon deeper, we also explore it from the perspective of data manifold and information.

**Data Manifold**: We show the distributions of synthetic images learned by the original baseline and baseline with discarding easy samples in Figure 1 (Middle) in the main text. Here we show all the distributions of baseline method, baseline with discarding hard samples, baseline with discarding easy samples, and baseline with the proposed ISA method in Figure 6. When the hard samples are discarded, it could be observed that the orange stars clusters together and the overlap with original dataset decreases. In contrast, the overlap grows when drop some easy samples, thereby depicting a better representation of the manifold. We also observe that compared to other cases, the application of ISA can help to make the stars more evenly distributed. This may explain why the generalization ability can be improved by ISA.

**Information**: To find out whether the harder samples have contained information in easier samples, we divide the target data into two splits: 80% samples that hold greater MSE loss in Eq. 4 and the left 20% easier ones. By treating the former as $X_{\mathcal{S}}$ and the latter as $X_{\mathcal{T}}$, we use the harder ones to make predictions for easier ones with Eq. 4. The MSE loss is 0.0661, as shown in Figure 1(Right). This loss indicates that the information in harder samples is enough for making precise predictions for easier samples. In contrast, when we use the 80% samples with smallest MSE loss to predict the left 20% hard ones, the loss is 0.1430. Therefore, harder samples cannot be replaced by simple samples.

**Diversity**: We also step further to compare the diversity of synthetic images with the recall value, which measures the expected likelihood of real samples against the synthetic manifold and is a commonly used metric in generative tasks for evaluating the diversity of generative model (Sajjadi et al., 2018). To be specific, in the generative model field, recall measures how much of a reference

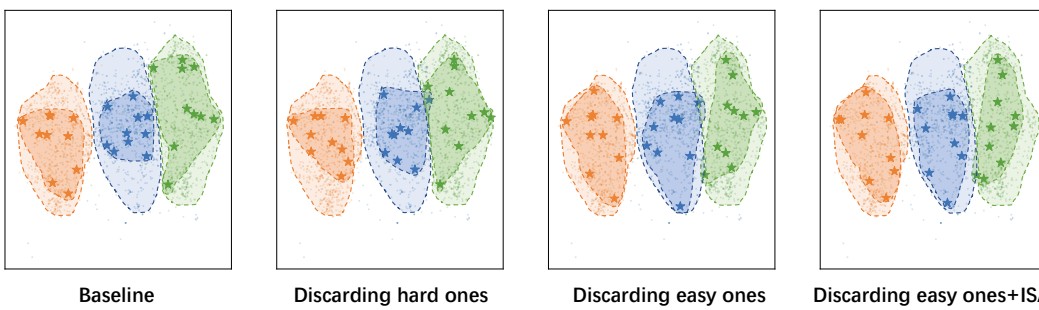

Figure 6: Distributions of synthetic images learned by different methods. The orange, blue, green points are the real images of three classes while the stars are the corresponding learned synthetic images. The orange stars clusters together and the overlap with original dataset decreases after discarding hard samples. In contrast, the overlap grows after discarding some easy samples, thereby depicting a better representation of the manifold. We also observe that compared to other cases, the application of ISA can help to make the stars more evenly distributed.

distribution can be generated by a part of a new distribution. Formally,

$$\text{recall} := \frac{1}{N} \sum_{i=1}^{N} 1_{R_i \in \text{manifold}(F_1, ..., F_M)}, \tag{8}$$

where N and M are the number of real and fake samples. $1_{(\cdot)}$ is the indicator function. $F_i$ is the $i$-th fake sample while $R_i$ is the $i$-th real sample. Manifolds are usually defined as:

$$\text{manifold}(R_1, ..., R_N) := \cup_{i=1}^{N} B(R_i, \text{NND}_k(R_i)), \tag{9}$$

where $B(x, r)$ is the sphere in $\mathbb{R}^D$ around $x$ with radius $r$. $\text{NND}_k(R_i)$ denotes the distance from $R_i$ to the $k$-th nearest neighbour among $\{R_i\}$ excluding itself.

To this end, the recall counts how many real samples occurs in the k-nearest neighbors of fake samples. We set $k = 5$. By treating the synthetic samples and original samples as fake and real samples respectively, we can calculate how many original samples can be recalled by generated images. A greater diversity in synthetic samples should recall more original samples. In other words, higher recall indicates greater diversity. With Eq. 8, we find that after incorporating the process of discarding easier samples in the outer loop, the recall value notably grows from 0.84 to 0.89. This improvement suggests an enhanced diversity in the synthetic samples with discarding easy samples.

## C  ALGORITHM ILLUSTRATION.

This paper introduces a dataset distillation algorithm based on crucial samples, which aims to distill a given labeled dataset into a smaller one so that a model trained on the small synthetic dataset can have a similar performance to the one trained on the original dataset, as shown in the right part of Figure 7. To achieve this goal, we first show that reducing redundancy in easy samples that are easy to be represented by the generated samples and taking more crucial samples into consideration can be beneficial for improving the diversity of synthetic samples and better depicting the data manifold in the dataset distillation tasks. Based on this observation, we further develop an infinite semantic augmentation-based dataset distillation algorithm, which takes an infinite number of virtual crucial samples into consideration in the semantic space. Through detailed mathematical analysis, the joint contribution to training loss of all interpolated feature points is formed into an analytical closed-form solution of an integral that can be optimized with almost no extra computational cost. As shown in Figure 7, given two input samples $(\mathbf{x}_1, \mathbf{y}_1)$ and $(\mathbf{x}_2, \mathbf{y}_2)$, we first extract their features $\mathbf{z}_1, \mathbf{z}_2$ and then adopt the loss in this figure to take all the interpolated points between them into consideration. $\delta_z = \mathbf{z}_1 - \mathbf{z}_2, \delta_y = \mathbf{y}_1 - \mathbf{y}_2$.

The whole algorithm can also be found in Algorithm 1. It is established based on a state-of-the-art pipeline FRePo (Zhou et al., 2022), which implements the dataset distillation by: sampling a model

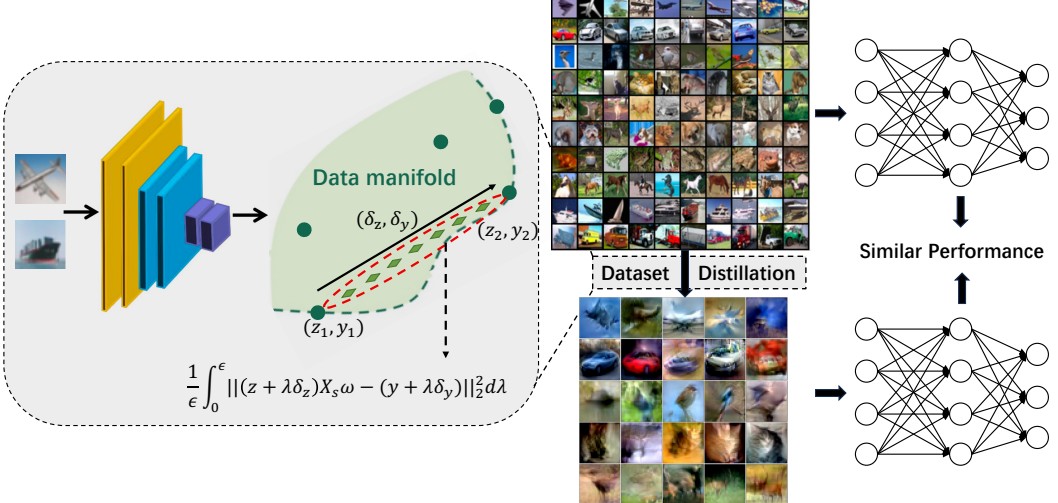

Figure 7: Dataset distillation with the assistance of crucial samples.

uniformly from a model pool $\mathcal{M}$ (Line 3) and a target batch $(X_\mathcal{T}, Y_\mathcal{T})$ uniformly from the labeled dataset $\mathcal{T}$ (Line 4), then computing the meta-training loss $\mathcal{L}$ (Line 5) to update the distilled data $\mathcal{S}$ (Line 8) (outer loop) and training the model $\theta_i$ on $\mathcal{S}$ (Line 9) (inner loop). To conduct the crucial samples based dataset distillation, we add the crucial sample exploring procedure by finding the top $p * 100\%$ percent images with the greatest meta-training loss (Line 6). Based on these samples, we further take more virtual samples into consideration via the new meta-learning loss (Line 7,8).

## D    INFINITE SEMANTIC AUGMENTATION

As the proposed Infinite Semantic Augmentation (ISA) takes an infinite number of virtual samples into consideration, one may be curious about whether the ISA will require more training steps for convergence. Figure 8 gives the answer. It suggests that there exists no big difference between the number of training steps for convergence of the baseline with that of method with ISA. Besides, the proposed method can achieve a better performance at the very early stages of training, indicating that the proposed method requires less time for a comparable performance. As for the training time cost, it is 2.5 hours (500,000 steps in total) under CIFAR10, IPC=10 setting while it is 2.4 hours for our baseline, indicating that the proposed module introduces negligible extra computational and time costs.

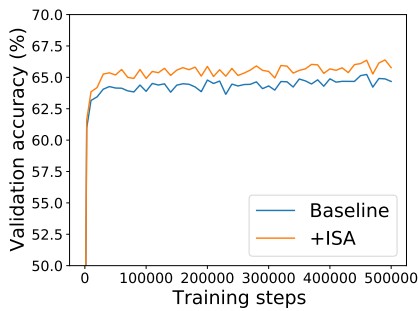

Figure 8: Validation accuracy during each training steps. It can be observed that adopting ISA will not require more training time for convergence.

The "single-step semantic augmentation" in Figure 3(c) indicates conducting a single-step mixup in feature space. As our method enables the augmentation of an infinite number of virtual samples in the semantic space by continuously interpolating between two target feature vectors, one may be curious about whether it is necessary to take infinite number of virtual samples into consideration. Therefore, in Figure 3(c), we conduct the single-step augmentation to take only one virtual samples between two samples into consideration during each step ( mixup in feature-space). It can be found that our ISA can hold a better performance against vanilla MixUp and single-step semantic augmentation, indicating the superiority of the proposed ISA. We will update our paper to improve the readability.

# E    IMPLEMENTATION DETAILS

All of our experiments are performed on a single NVIDIA A100 GPU with 80GB of GPU memory. We implement our method in JAX and reproduce previous methods using their officially released code. All the hyper-parameters are set following the released instructions. The training time for experiments on ImageNet including ImageNet-64, ImageNette, ImageWoof is around a week on a single A100, the same with the original ones. Other experiments only require several hours for training.

We also report the KRR predictors test accuracy using the feature extractor trained on the distilled data following FRePo (Zhou et al., 2022), which means obtaining the prediction with $K^\theta_{X_\mathcal{T} X_\mathcal{S}}(K^\theta_{X_\mathcal{S} X_\mathcal{S}} + \lambda I)^{-1} Y_\mathcal{S}$ in Eq. 4. With the KRR, we can achieve a higher test performance as shown in Table 67. The "ori" indicates training the neural networks with the distilled data and making predictions with the trained network, which is the default setting. However, we find the KRR predictor may fail to improve the performance when the distillation task is tough. For example, the test performance drops from 8.0% to 6.7% on the ImageNet dataset, as shown in Table 7.

Table 6: Distillation performance in term of KRR predicted test accuracy (%).

| Method | | CIFAR10 | | | CIFAR100 | | | TinyImageNet | |
|---|---|---|---|---|---|---|---|---|---|
| | | 1 | 10 | 50 | 1 | 10 | 50 | 1 | 10 |
| FRePo | ori | 46.8±0.7 | 65.5±0.4 | 71.7±0.2 | 28.7±0.1 | 42.5±0.2 | 44.3±0.2 | 15.4±0.3 | 25.4±0.2 |
| | KRR | 47.9±0.6 | 68.0±0.2 | 74.4±0.1 | 32.3±0.1 | 44.9±0.2 | 43.0±0.3 | 19.1±0.3 | 26.5±0.1 |
| Ours | ori | 48.4±0.4 | 67.2±0.4 | 73.8±0.0 | 31.2±0.2 | 46.4±0.5 | **49.4±0.3** | 19.8±0.1 | 27.0±0.3 |
| | KRR | **50.5±0.7** | **69.0±0.4** | **75.6±0.1** | **38.0±0.1** | **48.4±0.4** | 48.0±0.2 | **23.4±0.4** | **28.1±0.2** |

Table 7: Distillation performance in term of KRR predicted test accuracy (%) on ImageNet subsets.

| Method | | ImageNette (128x128) | | ImageWoof (128x128) | | ImageNet (64x64) | |
|---|---|---|---|---|---|---|---|
| | | 1 | 10 | 1 | 10 | 1 | 2 |
| FRePo | ori | 48.1±0.7 | 66.5±0.8 | 26.7±0.6 | 42.2±0.9 | 7.5±0.3 | 9.7±0.2 |
| | KRR | **50.6±0.6** | 67.1±0.7 | 31.3±0.9 | 43.5±0.8 | 7.2±0.2 | 9.5±0.2 |
| Ours | ori | 49.6±0.6 | 67.8±0.3 | 30.8±0.5 | 43.8±0.6 | **8.0±0.2** | **10.7±0.1** |
| | KRR | 48.5±0.6 | **69.2±0.4** | **33.6±0.5** | **46.3±0.4** | 6.7±0.2 | 7.6±1.0 |

# F    EXPERIMENTS ON MATCHING-BASED METHODS

This paper mainly focuses on exploring what kind of target data is crucial for dataset distillation in the outer loop of the meta-learning-based methods based on the analysis of both the matching-based and meta-learning-based methods in the secondary paragraph in Introduction. With the exploration in Section 2.2, we introduce a selection+augmentation method that can be adopted during the outer-loop of meta-learning-based methods. Therefore, apart from FRePo (Zhou et al., 2022), we also combined the proposed modules with various state-of-the-art meta-learning-based methods including RFAD (Loo et al., 2022), FRePo (Zhou et al., 2022), RCIG (Loo et al., 2023) in our ablation studies. The results in Table 4 validate the effectiveness of the proposed method. To further explore whether the proposed module can benefit for matching-based methods, we conduct experiments on classical DM (Zhao & Bilen, 2023) and MTT (Cazenavette et al., 2022) methods in Table 8. The performances are improved in most cases, indicating the effectiveness of the proposed approach.

Table 8: Test accuracies of applying our module to matching-based methods. * indicates the results are reproduced with the officially released codes.

| Methods | CIFAR10 | | | Methods | CIFAR10 | | |
|---|---|---|---|---|---|---|---|
| | 1 | 10 | 50 | | 1 | 10 | 50 |
| DM* | 25.9±0.8 | **48.9±0.6** | 62.7±0.5 | MTT* | 46.3±0.8 | 65.2±0.5 | 71.6±0.2 |
| +Ours | **26.5±0.6** | 48.5±0.4 | **62.9±0.2** | +Ours | **57.9±0.6** | **65.4±0.6** | **72.9±0.2** |

# G    MORE COMPARISONS WITH OTHER METHODS

**Comparison with dataset selection method.**

In this paper, we propose to focus more on cru-
cial samples during training for a better dataset
distillation performance. However, one can
also select crucial samples beforehand and then
conduct dataset distillation. To investigate the
comparison of our online method with the of-
fline dataset selection, we compare our method
with a newly released study "Prune then dis-
till", which proposes to prune the dataset first
to select the most important samples and then
conduct dataset distillation. We also compare
our method with a novel method DREAM (Liu

Table 9: Test accuracy (%) comparison with dataset selection method.

| Method | CIFAR10 | | |
|---|---|---|---|
| | 1 | 10 | 50 |
| Prune the distill | 44.7±1.5 | 63.1±0.7 | 69.7±0.4 |
| MTT+Ours | **57.9±0.6** | **65.4±0.6** | **72.9±0.2** |
| DREAM | 51.1±0.3 | 69.4±0.4 | **74.8±0.1** |
| IDC+Ours | **58.6±0.9*** | **71.1±0.1*** | 73.8±0.0 |

et al., 2023), who proposes to do clustering to fetch representative samples first and then conduct
dataset distillation every certain iterations. As shown in Table 9, our method still shows superiority to
data selection methods. Note that since the "prune then distill" and DREAM adopt the distillation
method MTT (Cazenavette et al., 2022) and IDC Kim et al. (2022) as their base methods, here we also
provide MTT+Ours and IDC+Ours* for a fair comparison. We think we hold two advantages. Firstly,
compared to the vallina dataset distillation methods, the offline pruning from "prune then distill"
will introduce extra time costs. In contrast, our method is an online one which just needs to rank
the losses and average the top-k losses, introducing almost no time costs as described in Section D.
Secondly, the offline pruning from "prune then distill" may result in information loss by dropping
samples before dataset distillation. Our online method ranks losses within batches during dataset
distillation, it is after the samples can be represented by the learned samples that these samples will
be discarded. In this way, the information in easy samples can be maintained.

We have also provide an example to compare our sampling strategy with DREAM. As shown in
Figure 9, when IPC=1, DREAM will get the distilled data (yellow circle) in the center of the each
class. This can lead to some unsuitable decision boundary (yellow line) learned by the distilled
data. In contrast, our method will push the distilled data to more hard-to-be-distinguished samples
that usually exist along the best decision boundary. In this way, the decision boundary (green lines)
learned by our distilled data (green circle) can be more precise. Besides, we have also showed an
example of the distilled data under multi-class classification tasks in Figure 10. Though it is hard to
draw a certain decision boundary, it can be observed that the distilled data made by focusing more on
hard samples can reach closer to the best decision boundary.

**Comparison to other state-of-the-art method.** Due to the
space limit, we only provide comparisons to a few state-of-
the-art methods in the main text. Here we also make more
comparisons in Table 11 and  12 with other recently released
methods including CAFE (Wang et al., 2022), FTD (Du et al.,
2023) and TESLA (Cui et al., 2023). Different from both
matching-based or meta-learning-based methods, IDC (Kim
et al., 2022) proposes a novel condensation framework that
generates multiple synthetic data with a limited storage budget

Table 10: Comparison with IDC.

| Method | CIFAR10 | | |
|---|---|---|---|
| | 1 | 10 | 50 |
| IDC | 36.7 | 58.3 | 69.5 |
| Ours | **48.4** | **67.2** | **73.8** |

via efficient parameterization considering data regularity. It analyzes the shortcomings of the existing
gradient matching-based condensation methods and develop an effective optimization technique for
improving the condensation of training data information. The comparsions are listed in Table 10.
Besides, it also proposes to divide images into several parts to make full use of storing budget,
making it a new state-of-the-art. We also include this strategy to make comparisons with IDC.
Our performances boost from 48.4%, 67.2% to 58.6%, 71.1% on CIFAR10, IPC=1, 10 while the
performances of IDC boost from 36.7%, 58.3% to 50.6%, 67.5%.

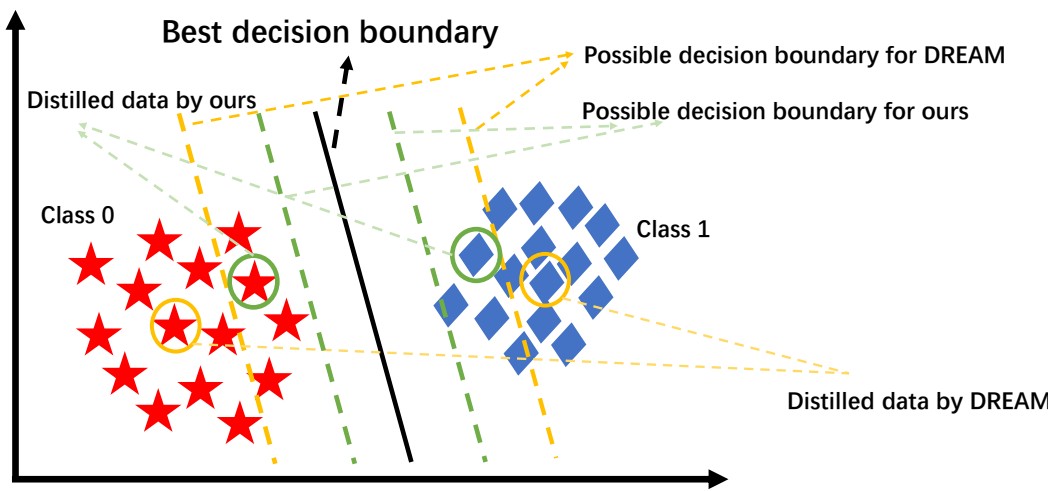

Figure 9: Method comparison with DREAM (Liu et al., 2023).

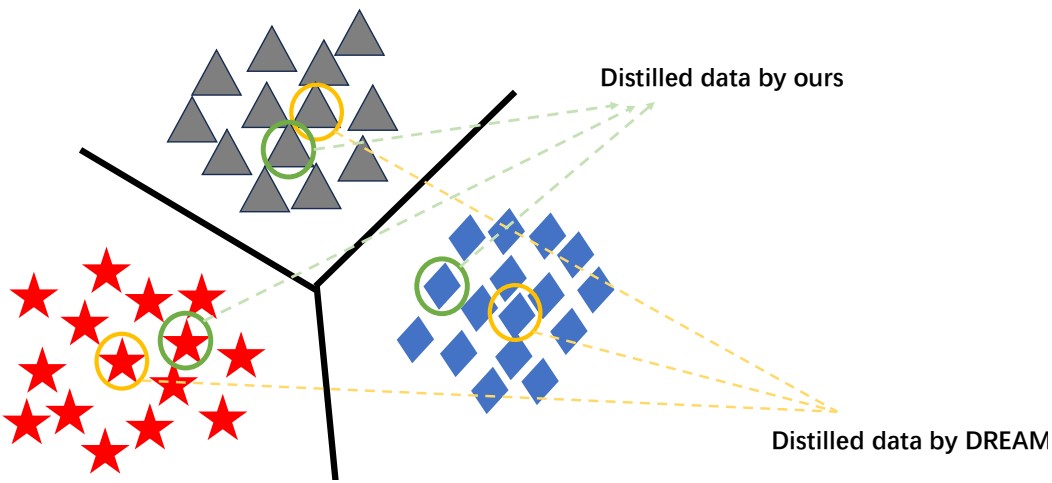

Figure 10: Method comparison with DREAM (Liu et al., 2023) (multi-class).

Table 11: Test accuracy (%) comparisons.

| Method | CIFAR10 | | | CIFAR-100 | | | T-ImageNet | |
|---|---|---|---|---|---|---|---|---|
| | 1 | 10 | 50 | 1 | 10 | 50 | 1 | 10 |
| CAFE | 31.6±0.8 | 50.9±0.5 | 62.3±0.4 | 14.0±0.3 | 31.5±0.2 | 42.9±0.2 | - | - |
| FTD | 46.8±0.3 | 66.6±0.3 | **73.8±0.2** | 25.2±0.2 | 43.4±0.3 | **50.7±0.3** | 10.4±0.3 | 24.5±0.2 |
| TESLA | **48.5±0.8** | 66.4±0.8 | 72.6±0.7 | 24.8±0.4 | 41.7±0.3 | 47.9±0.3 | - | - |
| Ours | 48.4±0.4 | **67.2±0.4** | **73.8±0.0** | **31.2±0.2** | **46.4±0.5** | 49.4±0.3 | **19.8±0.1** | **27.0±0.3** |

Table 12: Test accuracy (%) comparisons.

| Method | ImageNette(128x128) | | ImageWoof(128x128) | | ImageNet(64x64) | |
|---|---|---|---|---|---|---|
| | 1 | 10 | 1 | 10 | 1 | 2 |
| FTD (Du et al., 2023) | **52.2±1.0** | 67.7±0.7 | 30.1±1.0 | 38.8±1.4 | - | - |
| TESLA (Cui et al., 2023) | - | - | - | - | 7.7±0.2 | 10.5±0.3 |
| Ours | 49.6±0.6 | **67.8±0.3** | **30.8±0.5** | **43.8±0.6** | **8.0±0.2** | **10.7±0.1** |

# H  VISUALIZATION OF DISTILLED SAMPLES

In this section, we show our distilled samples of various datasets under different IPCs.

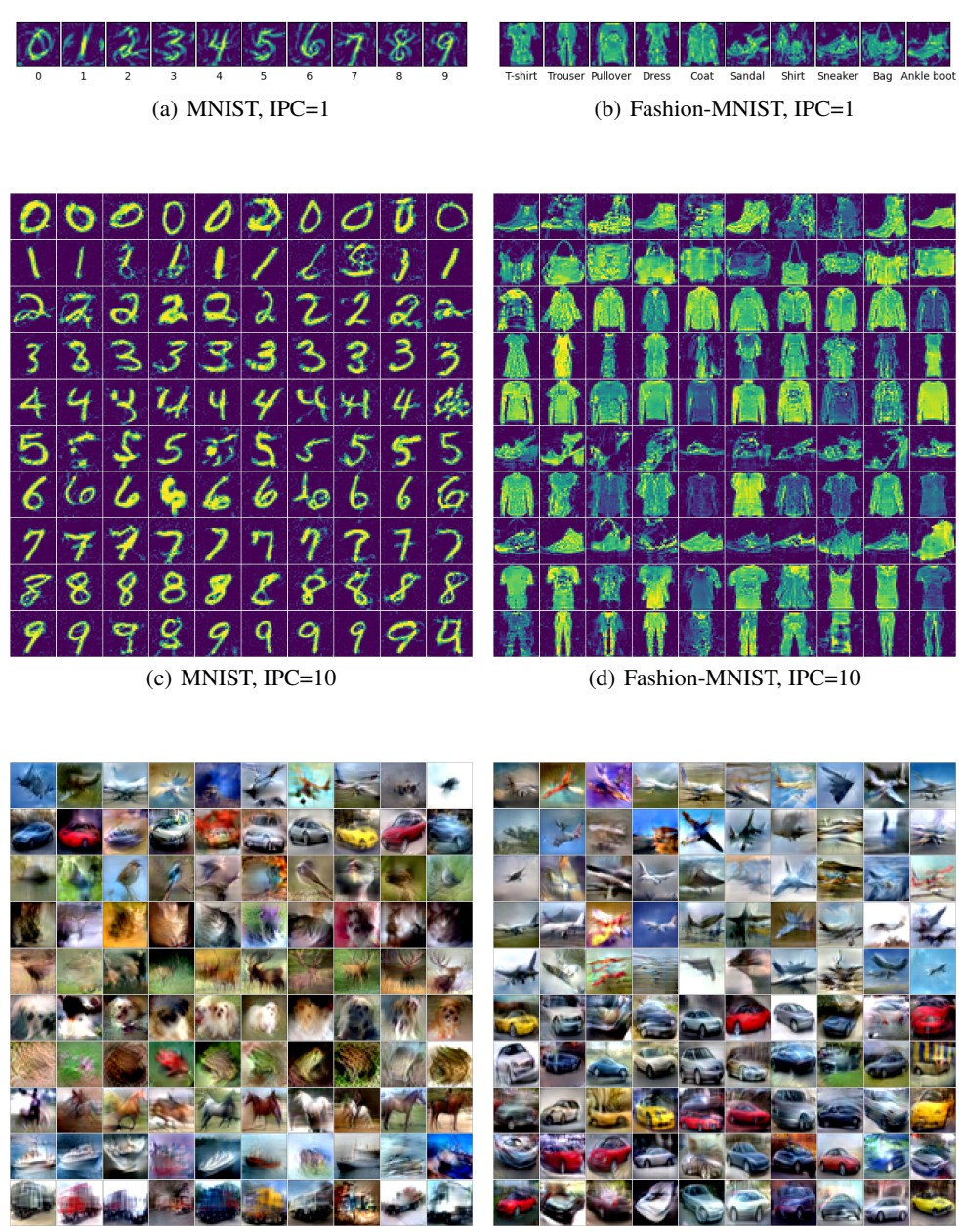

(a) MNIST, IPC=1

(b) Fashion-MNIST, IPC=1

(c) MNIST, IPC=10

(d) Fashion-MNIST, IPC=10

(e) CIFAR10, IPC=10

(f) CIFAR10, IPC=50

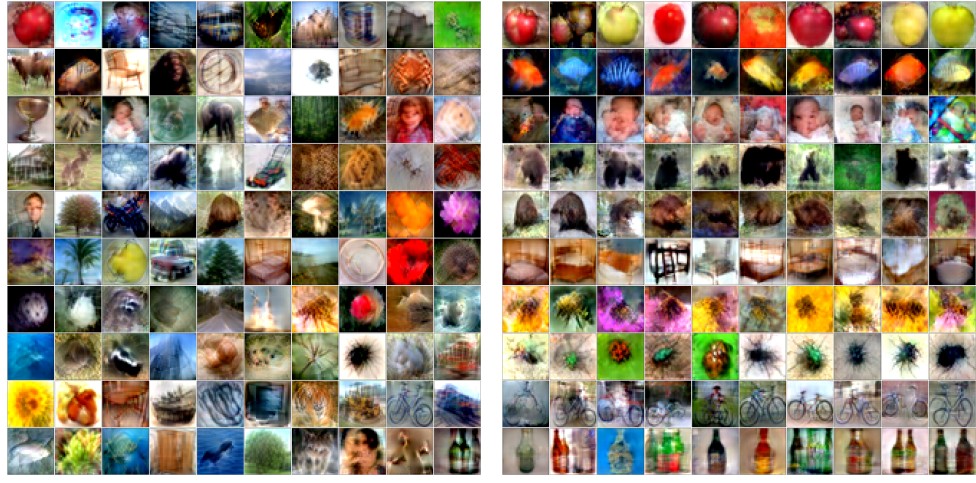

(g) CIFAR100, IPC=1    (h) CIFAR100, IPC=10

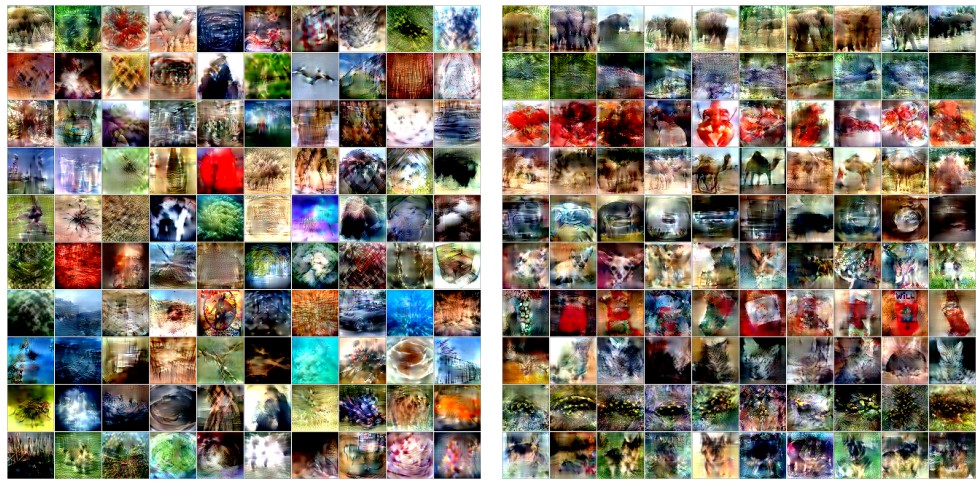

(i) TinyImageNet, IPC=1    (j) TinyImageNet, IPC=10

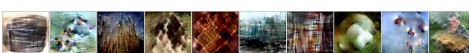    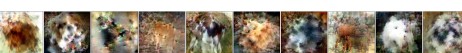

(k) ImageNette, IPC=1    (l) ImageWoof, IPC=1

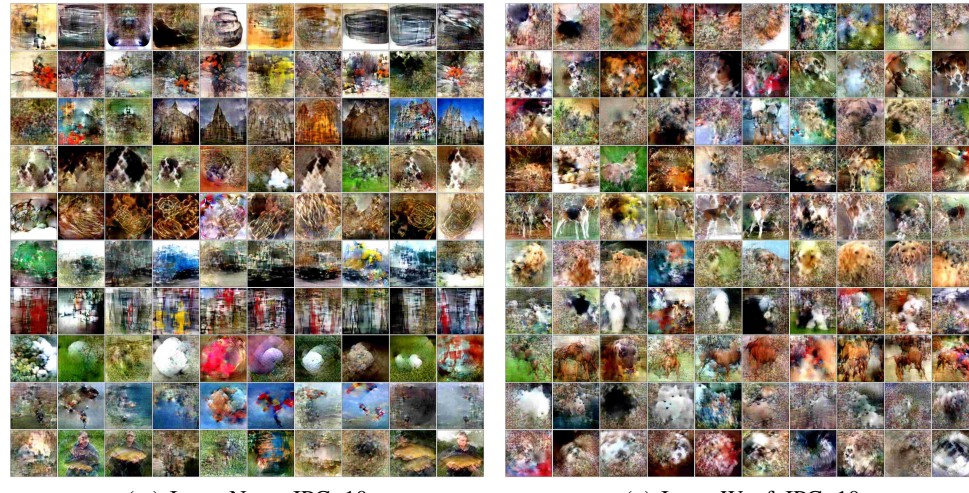

(m) ImageNette, IPC=10                    (n) ImageWoof, IPC=10

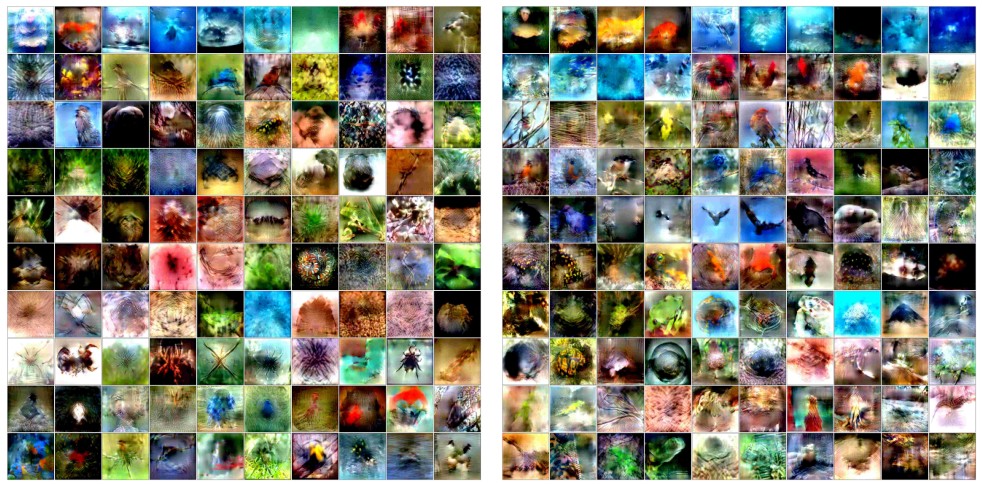

(o) ImageNet (64x64), IPC=1               (p) ImageNet (64x64), IPC=2

