# OpenReview forum: "Boosting Dataset Distillation with the Assistance of Crucial Samples"
_ICLR.cc/2024/Conference — Submitted to ICLR 2024_

### Official Review · Reviewer_dBFm · 2023-10-31

**Soundness:** 3 good
**Presentation:** 2 fair
**Contribution:** 3 good
**Rating:** 6
**Confidence:** 4

**Summary:**

The authors delve into the task of dataset distillation from the perspective of sample cruciality, they argue that hard samples in the original dataset contain more information. To this end, they discard some easier samples and enrich harder ones in the semantic space through continuously interpolating between two target feature vectors during data distillation.

**Strengths:**

The authors delve into the task of dataset distillation from the perspective of sample cruciality and propose the idea of adjusting the proportion of difficult and easy samples in the data distillation process; few papers considered this aspect before.
They put forward an infinite semantic augmentation method by continuously interpolating between two target feature vectors, requiring no extra computational costs while being effective.
The applicability of distilled data is considered, They demonstrated that their distilled data is capable of providing benefits to continual learning and membership inference defense.

**Weaknesses:**

The author only demonstrated through some simple experiments that in data distillation, the importance of difficult samples is stronger than that of simple samples. However, this conclusion cannot adequately explain that, in Figure 5 of the appendix, it can be observed that discarding difficult samples still allows the distilled data to achieve the comparable performance as the original distillation method, or even better.

Can the author provide more profound and solid explanations for the point mentioned above?

When comparing with baseline methods, the author did not compare with the state-of-the-art methods like FTD and TESLA. Nevertheless, this method only achieved state-of-the-art performance in 8 out of the 14 experimental setups.

Can the author complete the relevant comparative experiments and provide an objective analysis of the experimental results？


Directly using MSELoss as a criterion to discard a substantial proportion of samples.  Could this lead to a bias in distilled data?

**Questions:**

Please refer to the Weakness part.

---

> ### Author Response · Authors · 2023-11-16
> **Response to Reviewer dBFm  (Part 1/2)**
>
> Thank you for the time and questions. Here we answer your questions point-by-point as follows.
>
> >**Q1**: More explanations of discarding samples.
>
> **A1**: Thank you for the question. Actually, we try to explore "whether we should make a compromise to focus on the majority of data points around the center to ensure good performances on these data and what would happen if we discard these samples" (the last line in the first paragraph of Section 2.2). Motivated by this curiosity,  we conduct extensive experiments on discarding easy samples or hard samples in Section2.2 (both the Figure 1 and Figure 5). The results on three datasets and four settings all suggest that dropping easy samples helps dataset distillation in all cases. And when the IPC is small, dropping hard ones can also help sometimes, which is consistent with your guess. Here, we provide two explanations and hope it can solve your puzzles.
>  - Recall that the goal for dataset distillation is to condense the large dataset into a smaller one such that a model trained on it can achieve a comparable test performance as one trained on the original dataset (see Introduction). Namely, we need to depict a good decision boundary for classification based on the distilled data. However,  relying on easy samples solely may result in short-cut learning[a]. In contrast, hard samples are hard-to-be-distinguished samples (Eq.4) that usually exist along the decision boundary and thus can help to support a more explicit decision boundary. We think that's why hard samples can help to improve the dataset distillation performances.
> - We have also explored it from both the perspective of data manifold and information in the third paragraph of Section 2.2. In short, we observed an increased manifold overlap between the generated images and the original dataset after discarding some easy samples (Figure 1 middle two images), thereby depicting a better representation of the manifold. We also find that making precise predictions for simple samples with hard samples is easy but the reverse is not. This indicates that harder samples are more informative than easier ones.
>
> We have included this discussion in the Section B of the revision. We think this is a really interesting finding that makes our work unique and distinct. We hope it can bring more insights to scholars in dataset distillation.
>
> ***
>
> >**Q2**: Performance analysis.
>
> **A2**: In Table 1 and 3, we have conducted experiments on 8 datasets under 20 settings. Compared to our baseline, our proposed method improves the baseline with an average relative improvement of 6.6% and an absolute improvement of 2.0% on 16 settings. When compared to SOTA approaches, our method has achieved the SOTA performance under 14 settings. Besides, the performances can be further boosted with more advanced studies, as shown in Table 4 and 8. For example, when combined with RFAD, our performance on CIFAR10, IPC=1 can achieve performance up to 54.8\%, which is a new SOTA in Table 1.
>
> What's more, we find the proposed module only fails to show superiority on small datasets like MNIST and F-MNIST but still holds a comparable performance. In contrast, it holds a superior performance on larger datasets such as T-ImageNet and ImageNet as shown in Table 1 and 3. Since the dataset distillation aims to solve the difficulty in processing large datasets, we believe performing better on large datasets is more important nowadays.

---

> > ### Comment · Reviewer_dBFm · 2023-11-20
> >
> > Thanks for the reply. I have the further questions about this submission.
> >
> > 1. what are the key differences between the proposed crucial samples and the matching samples in the DREAM paper? How about comparing this method to DREAM?
> > 2. Discarding easy and hard, is there need to set a threshold? if so, hope the authors can show more details about it and explain whether this operation influences the generality of this method.
> >
> >
> >
> >
> >
> > DREAM: Efficient Dataset Distillation by Representative Matching. ICCV2023

---

> > > ### Author Response · Authors · 2023-11-20
> > > **Response to Reviewer dBFm**
> > >
> > > Thank you for the response. Here we answer your further questions point-by-point as follows.
> > >
> > > >**Q5**: Sample strategy compared to DREAM.
> > >
> > > **A5**: Our sampling strategy differs from DREAM from the aspects of follows:
> > > -  Cluster center vs. Harder samples: DREAM is a method that mainly proposes to conduct a clustering process (K-Means) before dataset distillation. In comparison, we rank the losses online within batches during dataset distillation. It is after the samples can be represented by the learned samples (loss in Eq.4 is small) that these samples will be discarded. In this way, the information in discarded samples can be maintained.
> > > -  Clustering vs. Boundary depicting: While both ways can improve the dataset distillation performance, the clustering operation of DREAM is developed based on the idea of distribution. In contrast, our method focuses more on depicting a better decision boundary as discussed in the response of A1.
> > > - To illustrate, we have provided an example in Figure 9 where we take a binary classification task as an example. When IPC=1, DREAM will get the distilled data (yellow circle) in the center of each class. This can result in an unsuitable decision boundary (yellow line) learned by the distilled data. In contrast, our method will push the distilled data to more hard-to-be-distinguished samples that usually exist along the best decision boundary. In this way, the decision boundary (green lines) learned by our distilled data (green circle) can be more precise.
> > > - Performance comparison: We provide the comparisons in the following table. Our results are produced by combining the proposed sample strategy with DREAM's baseline method for a fair comparison. Due to the training time cost of the baseline method (for example, it takes more than three days to conduct the experiments of CIFAR100, IPC=10), we can only provide results on small IPC right now.  More experiments will be included in the final revision when they are done. We have also included the above discussion in the updated manuscript.
> > >
> > > |       | CIFAR10, IPC=1 | CIFAR10, IPC=10 | CIFAR100, IPC=1 |
> > > |-------|----------------|-----------------|-----------------|
> > > | DREAM | 51.1±0.3       | 69.4±0.4        | 29.5±0.3        |
> > > | Ours  | **58.6±0.9**   | **71.1±0.1**    | **35.0±0.3**    |
> > >
> > > ***
> > >
> > > >**Q6**: Discarding threshold.
> > >
> > > **A6**:  Yes. The experiments about the threshold can be found in Section B and Figure 5. It suggests that discarding easier samples at a small rate can help for dataset distillation.  In this paper, the threshold is set to 0.2 in all experiments as illustrated in our Implementation Details (the first paragraph of Section 3).

---

> > > > ### Comment · Reviewer_dBFm · 2023-11-21
> > > >
> > > > DREAM is also an online sampling strategy, and the clustering process is conducted through the entire distillation process. DREAM's clustering process also utilizes the complete information of the class.
> > > > Under the binary classification, your concern makes sense. But if it is extended to multi-class classification, can it still describe better decision boundaries?  I am concerned that decision boundaries will appear in multiple directions, making it more difficult for the model to converge and lose its generalization ability.

---

> > > > > ### Author Response · Authors · 2023-11-21
> > > > > **Response to Reviewer dBFm**
> > > > >
> > > > > Thank you for the response.
> > > > >
> > > > > 1. DREAM: Yes, DREAM is also an online method and every certain iterations it will conduct a clustering process to fetch the representative samples before dataset distillation. The main difference lies in what we keep during sampling. That's why we use "Cluster center vs. Harder samples" as the sub-title instead of "Offline vs. Online". We have updated the revision to make the description of DREAM more detailed and rigorous.
> > > > >
> > > > > 2. Classification task: It makes sense that focusing more on harder samples makes it difficult for the model to converge. This is consistent with our results of Figure 5. Therefore, a proper threshold is required. And that's why we use a small discarding rate as described in A6. Besides, the results in Figure 3.a have showed the effectiveness of discarding operation on three datasets. Here we also provide more results on larger datasets to show its effectiveness. The number of classes ranges from 10 to 1000. We hope this can solve your concerns about the performance on multi-class classification.
> > > > >
> > > > > | Method      | ImageNette(128x128) |              | ImageWoof(128x128) |              | ImageNet(64x64) |              |
> > > > > |-------------|---------------------|--------------|--------------------|--------------|-----------------|--------------|
> > > > > |             | 1                   | 10           | 1                  | 10           | 1               | 2            |
> > > > > | Baseline    | 48.1±0.7            | 66.5±0.8     | **29.7±0.6**       | 42.2±0.9     | 7.5±0.3         | 9.7±0.2      |
> > > > > | +Discarding | **49.1±0.7**        | **66.6±0.4** | 28.0-0.4           | **43.0±0.7** | **8.6±0.2**     | **10.6±0.2** |

---

> > > > > > ### Comment · Reviewer_dBFm · 2023-11-21
> > > > > >
> > > > > > Thanks for your clarification. I will increase the score to 6.

---

> > > > > > > ### Author Response · Authors · 2023-11-21
> > > > > > > **Thanks for the recognition**
> > > > > > >
> > > > > > > Thanks a lot for your precious time in helping us improve our paper. We are encouraged by your recognition.
> > > > > > >
> > > > > > > Best regards,
> > > > > > >
> > > > > > > Authors

---

> ### Author Response · Authors · 2023-11-16
> **Response to Reviewer dBFm  (Part 2/2)**
>
> >**Q3**: Comparison with SOTA methods like FTD and TESLA.
>
> **A3**: The comparisons with FTD and TESLA are listed as follows. While these two methods are novel and provide good performances, it can be observed that our method holds a better performance in most cases. We have included this in Appendix G.
>
> | Method | CIFAR10      |              |              | CIFAR-100    |              |              | T-ImageNet   |              | ImageNette   |              | ImageWoof    |              | ImageNet    |              |
> |--------|--------------|--------------|--------------|--------------|--------------|--------------|--------------|--------------|--------------|--------------|--------------|--------------|-------------|--------------|
> |        | IPC=1        | IPC=10       | IPC=50       | IPC=1        | IPC=10       | IPC=50       | IPC=1        | IPC=10       | IPC=1        | IPC=10       | IPC=1        | IPC=10       | IPC=1       | IPC=2        |
> | FTD    | 46.8±0.3     | 66.6±0.3     | 73.8±0.2     | 25.2±0.2     | 43.4±0.3     | **50.7±0.3** | 10.4±0.3     | 24.5±0.2     | **52.2±1.0** | 67.7±0.7     | 30.1±1.0     | 38.8±1.4     | -           | -            |
> | TESLA  | **48.5±0.8** | 66.4±0.8     | 72.6±0.7     | 24.8±0.4     | 41.7±0.3     | 47.9±0.3     | -            | -            | -            | -            | -            | -            | 7.7±0.2     | 10.5±0.3     |
> | Ours   | 48.4±0.4     | **67.2±0.4** | **73.8±0.0** | **31.2±0.2** | **46.4±0.5** | 49.4±0.3     | **19.8±0.1** | **27.0±0.3** | 49.6±0.6     | **67.8±0.3** | **30.8±0.5** | **43.8±0.6** | **8.0±0.2** | **10.7±0.1** |
>
> ***
>
> >**Q4**: Could MSELoss lead to a bias in distilled data?
>
> **A4**: Discarding a substantial proportion of sample losses in Eq.4 can help the optimization focus more on the hard-to-be-distinguished samples, making the distilled data helpful/biased in depicting a better decision boundary so that a model trained on these distilled samples can show comparable performances to the one trained on original data. This is consistent with the goal of dataset distillation described in the Introduction: Dataset distillation aims to learn a small set of synthetic examples from a large dataset such that a model trained on it can achieve a comparable test performance as one trained on the original dataset.

---

> ### Author Response · Authors · 2023-11-20
> **Kind reminder to look at the authors' reply**
>
> Dear Reviewer dBFm:
>
> We thank you for the precious time and efforts in reviewing this paper. We have provided corresponding responses with elaborate discussions and extensive experiments on why dropping easy samples can help in dataset distillation, together with more discussion and comparisons with more studies. We have also include these in our revision (highlighted in blue). We hope to further discuss with you whether or not your concerns have been addressed appropriately. Please let us know if you have additional questions or ideas for improvement.
>
> Looking forward to your reply.
>
> Authors.

---

### Official Review · Reviewer_LMPF · 2023-10-31

**Soundness:** 3 good
**Presentation:** 3 good
**Contribution:** 3 good
**Rating:** 6
**Confidence:** 4

**Summary:**

The paper innovatively tackles the challenge of Dataset Distillation (DD) with a focus on sample cruciality in the outer loop of the bi-level learning problem. Building upon the neural Feature Regression (FRePo) framework, the authors introduce the Infinite Semantic Augmentation (ISA) algorithm. This algorithm enriches harder-to-represent samples in the semantic space through a process of continuous interpolation between two target feature vectors. Importantly, the algorithm is highly efficient as it formulates the joint contribution to training loss as an analytical closed-form integral solution. The method is rigorously evaluated on five benchmark datasets including MNIST, Fashion-MNIST, CIFAR10, CIFAR100, and Tiny-ImageNet. It is also compared against six baseline dataset distillation algorithms: DSA, DM, MTT, KIP, RFAD and FRePo. The experimental results demonstrate that the proposed ISA method effectively reduces dataset size while maintaining or even enhancing model accuracy. The distilled data also proves to be beneficial for downstream applications such as continual learning and privacy protection.

**Strengths:**

- **Originality**: The paper's focus on optimizing the outer loop in the bi-level optimization problem for Dataset Distillation is original.
- **Quality**: The paper is methodologically sound, demonstrated by a comprehensive set of experiments across five benchmark datasets. It also includes an ablation study that pinpoints the contributions of different components. The derivation of the integral into an analytical closed-form solution makes the algorithm an efficient solution.
- **Clarity**: The paper is well-organized and the algorithmic steps are outlined in detail.
- **Significance**: The proposed method is efficient, achieving state-of-the-art results in dataset size reduction while maintaining or even improving model performance.

**Weaknesses:**

1. **Inconsistent and Marginal Gains in Test Accuracy**: The test accuracy of the proposed method doesn't consistently outperform existing techniques. When it does show an improvement, the margin is sometimes minimal.
2. **Incomplete Review of Related Work**: The paper falls short in its coverage of existing literature. The need for a more comprehensive review is also detailed in the "Questions" section below.
3. **Lack of Comparative Analysis with Data Selection Algorithms**: The experiments in the paper do not include comparisons with data selection algorithms, leaving a gap in understanding how the proposed method stacks up against these approaches.

**Questions:**

1. **Clarification on "Data Extension" Terminology**: The term "data extension" is unfamiliar and appears to be non-standard. Is it synonymous with commonly used terms like "data augmentation" or "data interpolation"? If not, what differentiates it, and why opt for this term?
2. **Major Revisions in Related Work Section Needed**: The section on related work requires substantial updates for completeness and context.
    1. **Coresets**: The paper cited is neither the seminal work nor the most recent in the field of coresets. It would be beneficial to include at least these two papers [1*] and [2*], and consider citing earlier foundational works they mention, perhaps in an appendix.
    2. **Dataset Distillation**: In addition to [2*], works like [3*] and [4*] are missing from both the discussion and comparison tables. Also, MTT, which is covered in the experiments, lacks mention in the related work section. Please include these papers in both the textual discussion and comparative evaluations.

[1*] Yang, Y., Kang, H. & Mirzasoleiman, B.. (2023). Towards Sustainable Learning: Coresets for Data-efficient Deep Learning. *Proceedings of the 40th International Conference on Machine Learning*, in *Proceedings of Machine Learning Research* 202:39314-39330 Available from https://proceedings.mlr.press/v202/yang23g.html.

[2*] Shin, S., Bae, H., Shin, D., Joo, W. & Moon, I.. (2023). Loss-Curvature Matching for Dataset Selection and Condensation. *Proceedings of The 26th International Conference on Artificial Intelligence and Statistics*, in *Proceedings of Machine Learning Research* 206:8606-8628 Available from https://proceedings.mlr.press/v206/shin23a.html.

[3*] Kim, J., Kim, J., Oh, S.J., Yun, S., Song, H., Jeong, J., Ha, J. & Song, H.O.. (2022). Dataset Condensation via Efficient Synthetic-Data Parameterization. *Proceedings of the 39th International Conference on Machine Learning*, in *Proceedings of Machine Learning Research* 162:11102-11118 Available from https://proceedings.mlr.press/v162/kim22c.html.

[4*] Wang, K., Zhao, B., Peng, X., Zhu, Z., Yang, S., Wang, S., ... & You, Y. (2022). Cafe: Learning to condense dataset by aligning features. In *Proceedings of the IEEE/CVF Conference on Computer Vision and Pattern Recognition* (pp. 12196-12205).

---

> ### Author Response · Authors · 2023-11-16
> **Response to Reviewer LMPF (Part 1/2)**
>
> Thank you for your positive feedback and questions. Here we answer your questions point-by-point as follows.
>
> >**Q1**: The test accuracy of the proposed method doesn't consistently outperform existing techniques. When it does show an improvement, the margin is sometimes minimal.
>
> **A1**: In Table 1 and 3, we have conducted experiments on $8$ datasets under $20$ settings. *Compared to our baseline*, our proposed method improves the baseline with an average relative improvement of $6.6$\% and an absolute improvement of $2.0$\% on $16$ settings, which is not marginal in dataset distillation field (can be inferred by comparisons with SOTA methods RFAD, MTT and FRePo in Table 1). *When compared to SOTA approaches*, our method has achieved the SOTA performance under $14$ settings. Besides, the performances can be further boosted with more advanced studies, as shown in Table 4 and 8. For example, when combined with RFAD, our performance on CIFAR10, IPC=1 can achieve performance up to $54.8$\%, which is a new SOTA in Table 1.
>
> What's more, we find the proposed module only fails to show superiority on small datasets like MNIST and F-MNIST but still shows to be comparable to other methods. In contrast, it holds a superior performance on larger datasets such as TinyImageNet and ImageNet as shown in Table 1 and 3. Since the dataset distillation aims to solve the difficulty in processing large datasets, we believe performing better on large datasets is more important nowadays.
>
> ***
>
> > **Q2**: Incomplete Review of Related Work
>
> **A2**: Thank you for the suggestion. In this paper, we mainly discussed studies like MTT(CVPR2022), FRePo(NeuraIPS2022), RCIG(ICML2023). We have also included the discussion of the suggested work [1*][2*][3*][4*] and experimental comparison with the dataset distillation methods [3*] (IDC) [4*] (CAFE) in Section G of the revision. CAFE proposes to conduct feature alignment for a better dataset distillation performance while IDC is a novel method that analyzes the shortcomings of the existing gradient matching-based condensation methods and develops an effective optimization technique for improving the condensation of training data information. The comparisons are listed in the table below (IDC only provides its mean accuracy in its Table 10, here we only show the mean accuracy without standard deviation value). More results can be found in the revision. Besides, IDC also proposes to divide images into several parts to make full use of the storage budget (IDC+M). We also include this strategy (Ours+M) to make comparisons with IDC in table below.
>
> Table a. Test accuracy (\%) comparison.
> | Method | CIFAR10  |          |          |
> |--------|----------|----------|----------|
> |        | IPC=1    | IPC=10   | IPC=50   |
> | CAFE   | 31.6     | 50.9     | 62.3     |
> | IDC    | 36.7     | 58.3     | 69.5     |
> | Ours   | **48.4** | **67.2** | **73.8** |
> | IDC+M    | 50.6     | 67.5     | **74.5**     |
> | Ours+M   | **58.6** | **71.1** | 74.2 |

---

> ### Author Response · Authors · 2023-11-16
> **Response to Reviewer LMPF (Part 2/2)**
>
> >**Q3**: Lack of Comparative analysis with data selection algorithms
>
> **A3**: Thank you for the question. In this paper, we try to explore what kind of data is more critical during dataset distillation, "whether we should make a compromise to focus on the majority of data points around the center to ensure good performances on these data and what would happen if we discard these samples" (the last line in the first paragraph of Section 2.2). Motivated by this curiosity,  we conducted extensive experiments on discarding easy samples or hard samples during training.
>
> Compared to traditional data selection algorithms that find crucial samples first, requiring extra processing time and may lose some information in removed samples, our online method ranks losses within batches during dataset distillation. It is after the samples can be represented by the learned samples (loss in Eq.4 is small) that these samples will be discarded. In this way, the information in discarded samples can be maintained. Besides, the time cost for sorting losses is negligible, making our method more efficient than offline data selection methods. We have also compared our method with 'prune then distill'[a] which prunes the dataset first and then conducts dataset distillation. The experimental results show that when both are combined with MTT, we can achieve a higher performance. Besides, we have also compared our method with DREAM, who is also online method and proposes to conduct clustering (K-Means) to fetch representative samples before dataset distillation every certain iterations. After combining the proposed sample strategy with DREAM's baseline method for a fair comparison, our method can achieve $58.3$\%(CIFAR10, IPC=1), $71.1$\%(CIFAR10, IPC=10), $35.0$\%(CIFAR100, IPC=1) while they are $51.1$\%, $69.4$\%, $29.5$\% for DREAM. We also provide a figure illustrating the sampling differences between DREAM and Ours in Figure 9 of the revision. We hope it can solve your concerns. More detailed discussions can be found in Section G of the revision.
>
> Table b. Test accuracy (\%) comparison.
> | Method             | CIFAR10      |              |              |
> |--------------------|--------------|--------------|--------------|
> |                    | IPC=1        | IPC=10       | IPC=50       |
> | Prune then distill | 44.7±1.5     | 63.1±0.7     | 69.7±0.4     |
> | Ours+MTT               | **57.9±0.6** | **65.4±0.6** | **72.9±0.2** |
>
>
> ***
>
> >**Q4**: Why Data extension and not data augmentation/interpolation
>
> **A4**: Thank you for the question. Both data augmentation and interpolation are techniques designed for dataset extension and differ from each other regarding the processing method. In other words, 'data extension' is the goal while 'data augmentation' and 'interpolation' are methods. In this paper, we named the title of Section 2.3 as 'Crucial sample extension with semantic augmentation' since we propose a method to "implicitly enrich harder ones in the semantic space through continuously
> interpolating between two target feature vectors" (the last paragraph of Introduction). We use semantic augmentation instead of data augmentation due to the reason that data augmentation typically refers to techniques that generate new training samples by applying various transformations or modifications to existing data, such as rotation, flipping, or adding noise. It is usually done before putting samples into models. In contrast, our proposed method conducts augmentation in semantic space. Experimental comparisons are also provided in Figure3.c, which indicates the superiority of the proposed semantic augmentation in the dataset distillation task.
>
> ***
> **References**:
>
> [a] Prune then distill: Dataset distillation with importance sampling, ICASSP 2023.
>
> [b] DREAM: Efficient Dataset Distillation by Representative Matching. ICCV2023.

---

> ### Author Response · Authors · 2023-11-22
> **Kind reminder to look at the authors' reply**
>
> Dear Reviewer LMPF:
>
> Thank you for the precious time and valuable suggestions to our paper. We have updated our revision to include your mentioned works and make comparisons with them, including more data selection methods and dataset distillation methods. And we also give the explanations on our performance gain in our last response. We hope these can address your concerns. As the rebuttal stage is drawing to a close,  we hope to further discuss with you whether or not your concerns have been addressed appropriately. Please let us know if you have additional questions or ideas for improvement.
>
> Looking forward to your reply.
>
> Authors.

---

> ### Author Response · Authors · 2023-11-23
> **Another kind reminder to look at the authors' reply**
>
> Dear reviewer LMPF:
>
> Sorry for bothering you again. Since there is **only a few hours left until the end of the discussion**, we hope to further discuss with you whether or not your concerns have been addressed appropriately by our responses. Please let us know if you have additional questions or ideas for improvement.
>
> Looking forward to your reply.
>
> Authors.

---

> > ### Comment · Reviewer_LMPF · 2023-11-23
> > **Thank you for your response**
> >
> > I want to thank the authors for their timely response. I'm still concerned about the inconsistency of the improvements, but all my other questions have been addressed by the authors. I will keep my initial rating of 6.

---

> > > ### Author Response · Authors · 2023-11-23
> > > **Response to Reviewer LMPF**
> > >
> > > Thank you for the response.
> > >
> > > As we previously mentioned, firstly, we have achieved SOTA in most scenarios (datasets including CIFAR10, CIFAR100, TinyImageNet, ImageNette, ImageWoof, ImageNet-1K) and have also verified that we can further enhance our performance by combining with the SOTA methods in Table 4 and Table 8. Since there is only one hour left until the end of the discussion phase, we are unable to provide experimental results in all scenarios immediately. However, the combination with RFAD has already allowed us to achieve new SOTA on CIFAR10 with IPC=1 as shown in Table 4. Therefore, we believe that we can also reach SOTA results on MNIST and Fashion-MNIST by combining with these methods, especially since we are already comparable to SOTA. We will also include these experiments once they are done in the revision. Moreover, we must reiterate our believe that the value of achieving good results on large datasets such as ImageNet far surpasses the value of achieving SOTA on smaller datasets like MNIST and Fashion-MNIST.

---

### Official Review · Reviewer_R7gb · 2023-10-31

**Soundness:** 4 excellent
**Presentation:** 4 excellent
**Contribution:** 2 fair
**Rating:** 6
**Confidence:** 5

**Summary:**

This paper proposes two techniques to boost the performance of kernel-based dataset distillation methods: discarding easy samples and infinite semantic augmentation. Experiments on several benchmarks demonstrate the effectiveness of the proposed methods.

**Strengths:**

1. The proposed infinite semantic augmentation technique is mathematically elegant and effective.
2. The experimental evaluations are comprehensive enough to validate the effectiveness.
3. The writing is coherent and easy to follow.

**Weaknesses:**

My major concern is on discarding easy samples.
1. On the one hand, this step requires computing the NFR loss in FRePo twice, which would require longer running time for dataset distillation and make the baseline complex.
2. On the other hand, this technique is heuristic and seems counterfactual. Intuitively, easy samples should contain some common patterns that can reflect what a class of objects looks like in general. These samples should be more effective than hard samples to capture the major features of each class. In dataset distillation, major information is expected to be stored while other unusual patterns are discarded. It seems strange to me that discarding easy samples leads to better performance, especially when IPC is small.
3. This strategy drops some samples, which destroys the original data distribution. However, according to Fig. 1, it makes the distilled data better follow the original distribution, which seems strange to me.
4. Moreover, there seems to be a paper using a similar technique [a].
5. I would like to see separate results of only using this strategy without ISA, such as in Tab. 2, 3, 4, and 8.
6. I suggest the authors compare qualitative samples of the baseline, with selection, and with ISA together to better reflect the functionality of each part. Currently it seems that the qualitative results are not different from the original FRePo too much.

[a] Prune then distill: Dataset distillation with importance sampling, ICASSP 2023.

**Questions:**

Please refer to Weaknesses for details.

---

> ### Author Response · Authors · 2023-11-16
> **Response to Reviewer R7gb (Part 1/3)**
>
> Thank you for your time and valuable questions. We answer your questions point-by-point as follows.
>
> >**Q1** : Discarding easy samples requires computing the NFR loss in FRePo twice, which would require longer running time for dataset distillation and make the baseline complex.
>
> **A1**: *The proposed method only requires to compute the 'NFR' loss once*. To be specific, we first calculate the 'NFR' loss with Eq.4 to get the loss of each sample. Afterward, we sort the losses and average those with greater losses to obtain the final loss. Hence, compared to the baseline, we only added a step of sorting, whose time complexity is $\mathcal{O}(B \log B)$. $B$ is the batch size. This time cost for ranking is negligible compared to the forward and backward time costs. The python-like pseudo-code relative to the 'NFR loss' is listed below.  Baseline is `    kernel_loss = mean_squared_loss(preds, labels).mean()  `, and ours is as follows:
>
> `   kernel_loss = mean_squared_loss(preds, labels)  `
> `   _, idx_ranked = top_k(kernel_loss, len(kernel_loss))  `
> `  kernel_loss = (kernel_loss[idx_picked[:int((1-discarding_rate))*len(idx_ranked))]]).mean()     `
>
> Furthermore, we have also provided the time cost comparison with our baseline in Appendix D, which writes "As for the training time cost, it is 2.5 hours (training time for 500,000 steps in total) under CIFAR10, IPC=10 setting while it is 2.4 hours for our baseline". This indicates that the proposed module introduces negligible extra computational and time costs.
>
> ***
>
> >**Q2** : Discarding easy samples is heuristic and seems counterfactual.
>
> **A2**: Thank you for the question . Actually, we hold the same curiosity with you on "whether we should make a compromise to focus on the majority of data points around the center to ensure good performances on these data and what would happen if we discard these samples" (the last line in the first paragraph of Section 2.2). Motivated by this curiosity,  we conduct extensive experiments on discarding easy samples or hard samples in Section2.2 (both the Figure 1 and Figure 5). The results on three datasets and four settings all suggest that dropping easy samples helps dataset distillation in all cases. Here we provide some explanations below.
> - The goal for dataset distillation is to condense the large dataset into a smaller one such that a model trained on it can  achieve a comparable test performance as one trained on original dataset (see Introduction). In other words, we need to depict a good decision boundary for classification based on the distilled data. However,  relying solely on easy samples may result in short-cut learning[a]. In contrast, hard samples are difficult-to-be-distinguished samples (Eq.4) that usually exist along the decision boundary thus can help to support a more explicit decision boundary. We think that's why hard samples can help to improve the dataset distillation performances.
> - We have also explored it from both the perspective of data manifold and information in the third paragraph of Section 2.2. In short, we observed an increased manifold overlap between the generated images and the original dataset after discarding some easy samples (Figure 1 middle two images), thereby depicting a better representation of the manifold. We also find that making precise predictions for simple samples with hard samples is easy but the reverse is not. This indicates that harder samples are more informative than easier ones.
>
> The above discussion has been included in Appendix B of the revision. We think this is a really interesting finding and it is the 'counterfactual nature' of our discovery that makes our work unique and distinct. We hope it can bring more insights to scholars in dataset distillation.
>
> ***
>
> >**Q3**: The data distribution in  Fig. 1.
>
> **A3**: In fact, we found that compared to the distilled samples generated by the baseline, the samples distilled by our method tend to be distributed not only in high-density spaces (central regions) but also in low-density spaces. This helps to increase the data manifold overlap with the original dataset.

---

> > ### Comment · Reviewer_R7gb · 2023-11-22
> >
> > I would like to thank the authors for the detailed response. I have understood that discarding easy samples may help DD generate more samples near decision boundaries, which can further help the learning of these boundaries. It would be better if there could be more theoretical analysis on this part.
> >
> > Overall, my concerns are alleviated and I choose to raise my score to 6.

---

> > > ### Author Response · Authors · 2023-11-22
> > > **Thanks to your recognition**
> > >
> > > Thank you a lot for your recognition. We are encouraged that our response have alleviated your concerns. We have also attached an example illustrating the differences of distilled data after focusing more on harder samples in Figure 9 of the revision. We hope it can help you further in understanding why hard samples can help.
> > >
> > > Again, thank you for the time and help in improving our paper.
> > >
> > > Best regards,
> > >
> > > Authors

---

> ### Author Response · Authors · 2023-11-16
> **Response to Reviewer R7gb (Part 2/3)**
>
> >**Q5**: Separate results of only using dropping strategy in Tab2, 3, 4, 8
>
> **A5**: We list the results of using dropping/discarding strategy of Table 2, 3, 4, 8 in the following:
> -  Table 2 evaluates the cross-architecture generalization ability of the distilled data. This evaluation with only discarding operation can be found in Figure 3.d. To be short, removing easy samples can result in a small generalization ability degradation. However, this can be alleviated by our proposed ISA, as shown in Table 2 and Figure 3.d.
>
> - Table 3 provides the evaluation of a larger dataset. The ablation studies of each proposed module on three datasets (CIFAR10, CIFAR100, Tiny-ImageNet) can be found in Figure 3.a. All the results validate the effectiveness of the proposed discarding strategy and ISA. We have also conducted experiments to provide discarding ablations on ImageNet datasets below. The results are consistent with ablation results in Figure 3.a that discarding some easy samples can help in dataset distillation.
>
> Table b: Test accuracy (\%) of adding discarding only (corresponding to Table 3 in main text)
> | Method      | ImageNette(128x128) |              | ImageWoof(128x128) |              | ImageNet(64x64) |              |
> |-------------|---------------------|--------------|--------------------|--------------|-----------------|--------------|
> |             | 1                   | 10           | 1                  | 10           | 1               | 2            |
> | Baseline    | 48.1±0.7            | 66.5±0.8     | **29.7±0.6**       | 42.2±0.9     | 7.5±0.3         | 9.7±0.2      |
> | +Discarding | **49.1±0.7**        | **66.6±0.4** | 28.0-0.4           | **43.0±0.7** | **8.6±0.2**     | **10.6±0.2** |
>
> - Table 4 and  Table 8 provide experimental results of combining the proposed modules with other methods. We list the required results of adding discarding operation only to these methods below. Discarding can boost the performances in most scenarios. This is consistent with our findings.
>
> Table c: Test accuracy (\%) of adding discarding only (corresponding to Table 4 in main text)
> | Method | CIFAR10  |              |              |              |          |              |
> |--------|----------|--------------|--------------|--------------|----------|--------------|
> |        | IPC=1    |              | IPC=10       |              | IPC=50   |              |
> |        | Baseline | +Discarding  | Normal       | +Discarding  | Normal   | +Discarding  |
> | RFAD   | 52.1±0.1 | **54.3±0.1** | 65.3±0.1     | **66.5±0.1** | 69.8±0.2 | **70.1±0.1** |
> | FRePo  | 46.8±0.7 | **48.0±0.4** | 65.5±0.4     | **66.8±0.4** | 71.7±0.2 | **72.9±0.1** |
> | RCIG   | 53.9±0.5 | **53.9±0.3** | 67.3±0.3     | **67.7±0.4** | 73.5±0.2 | **73.7±0.4** |
>
> Table d: Test accuracy (\%) of adding discarding only (corresponding to Table 8 in main text)
> | Method | CIFAR10  |              |              |              |          |              |
> |--------|----------|--------------|--------------|--------------|----------|--------------|
> |        | IPC=1    |              | IPC=10       |              | IPC=50   |              |
> |        | Baseline | +Discarding  | Normal       | +Discarding  | Normal   | +Discarding  |
> | DM     | 25.9±0.8 | **26.1±0.3** | **48.9±0.6** | 48.4±0.6     | 62.7±0.5 | **62.8±0.2** |
> | MTT    | 46.3±0.8 | **52.9±0.7** | 65.2±0.5     | **65.2±0.5** | 71.6±0.2 | **72.1±0.2** |

---

> ### Author Response · Authors · 2023-11-16
> **Response to Reviewer R7gb (Part 3/3)**
>
> >**Q4**: Comparison to 'Prune then distill'.
>
> **A4**: 'Prune then distill' is a method that proposes to prune the dataset first and then conduct dataset distillation. Our method differs from it from two perspective:
> - Compared to the vallina dataset distillation methods, the offline pruning will introduce extra time costs that you mentioned in the first question. In contrast, our method is an online one which just needs to rank the losses and average the topk losses, introducing almost no time costs as illustrated in the Answer to the fisrt question.
> - The offline pruning may result in information loss by dropping samples before dataset distillation. Our online method ranks losses within batches during dataset distillation, it is after the samples can be represented by the learned samples (*a.k.a* 'NFR' loss is small) that these samples will be discarded. In this way, the information in easy samples can be maintained. The experimental comparisons are listed as follows. Note that since the 'prune then distill' adopts the distillation method MTT as their distillation method, here we provide MTT+Ours for a fair comparison in the table below, which indicates that our proposed method has a better performace. Detailed information can be found in Section G in the revision.
>
> Table a: Test accuracy (\%) comparison
> | Method             | CIFAR10      |              |              |
> |--------------------|--------------|--------------|--------------|
> |                    | IPC=1        | IPC=10       | IPC=50       |
> | Prune then distill | 44.7±1.5     | 63.1±0.7     | 69.7±0.4     |
> | Ours               | **57.9±0.6** | **65.4±0.6** | **72.9±0.2** |
>
> ***
>
> >**Q6**: Ablation study and the performance gain.
>
> **A6**: The ablation study of each component is provided in the first paragraph (**Ablation Studies on Each Proposed Module.**) of Section 3.2 (ABLATION STUDIES). It is carried out on three datasets (Figure 3.a) and all the results indicate the effectiveness of both the proposed modules.
>
> As for the performance gain, in Table 1 and 3, we have conducted experiments on $8$ datasets under $20$ settings. Compared to the baseline, our proposed method improves the baseline with an average relative improvement of $6.6$\% and an absolute improvement of $2.0$\% on $16$ settings, which is not marginal in the dataset distillation field (can be inferred by comparisons with SOTA methods RFAD, MTT and FRePo in Table 1).

---

> ### Author Response · Authors · 2023-11-20
> **Kind reminder to look at the authors' reply**
>
> Dear Reviewer R7gb:
>
> We thank you for the precious review time and valuable comments. We have provided corresponding responses with elaborate discussions and extensive experiments on why dropping easy samples can help in dataset distillation,  which we hope to address your concerns. We hope to further discuss with you whether or not your concerns have been addressed appropriately. Please let us know if you have additional questions or ideas for improvement.
>
> Looking forward to your reply.
>
> Authors.

---

> ### Author Response · Authors · 2023-11-21
> **Another kind reminder to look at the authors' reply**
>
> Dear Reviewer R7gb:
>
> Sorry to bother you again. As the rebuttal stage is drawing to a close (only 1-2 days left), we wanted to bring to your attention that we have not yet received any feedback from you yet. We have provided corresponding responses with elaborate discussions and extensive experiments on why dropping easy samples can help in dataset distillation a few days ago, which we hope to address your concerns. We hope to further discuss with you whether or not your concerns have been addressed appropriately. Please let us know if you have additional questions or ideas for improvement.
>
> Looking forward to your reply.
>
> Authors.

---

### Official Review · Reviewer_wdbv · 2023-12-14

**Soundness:** 2 fair
**Presentation:** 3 good
**Contribution:** 2 fair
**Rating:** 5
**Confidence:** 4

**Summary:**

This paper introduces an Infinite Semantic Augmentation (ISA) method for dataset distillation, which enhances the performance of existing methods like MTT and IDC. The authors begin by demonstrating that discarding easy samples from the original datasets is beneficial, as it focuses on extracting crucial features from the more challenging samples. Drawing inspiration from MixUp, they propose an augmentation technique to enrich these difficult samples. The effectiveness of the proposed ISA is verified through experiments conducted on CIFAR, Tiny ImageNet, and ImageNet subsets.

**Strengths:**

Strength:
1. This method is an augmentation-based method that can be embedded with other existing dataset distillation methods for improved performance.

2. The authors evaluate the effectiveness of the proposed ISA in downstream tasks, including continual learning and membership inference defense.

3. The visualizations in this paper are impressive and comprehensive, significantly enhancing its clarity.

**Weaknesses:**

Weakness:
1. This paper offers limited new insights into dataset distillation. The first aspect of ISA suggests discarding easy samples to enhance dataset distillation, a concept already proposed by Dream[1] at ICCV 2023, emphasizing the importance of focusing on significant samples. The second aspect of ISA essentially adapts MixUp and CutMix techniques, commonly employed in training DNNs.

2. The performance improvement offered by ISA is marginal, and the baselines across different tables appear selectively chosen. ISA notably underperforms compared to Dream, especially evident in specific settings like CIFAR-100 ipc 50, where ISA achieves only a 49.4% success rate compared to Dream's 52.6%. The authors selectively report only those baselines and settings where ISA outperforms other methods, while neglecting strong baselines like IDC, Haba[2], and Sre2L[3], as well as significant settings like CIFAR-100 in Dream.

3. The paper's definition of 'hard' examples lacks sufficient detail and persuasiveness. The authors use Equation (4) as a metric to rank samples and eliminate 'easy' examples. However, they omit the derivation of Equation (4). Notably, Equation (4) closely resembles the baseline method KRR[4], yet the authors fail to provide any theoretical justification or derivation for this equation.

4. This paper is overclaimed. The authors claimed "Through detailed mathematical derivation" in abstract, but as stated in point 3, Section 2 lacks the derivation of Equation (4). The authors claimed "compare our method to several state-of-the-art dataset distillation methods", but they omitted the strong baselines as stated in point 2.

5. The paper's conclusion that 'discarding easy samples is beneficial' is based merely on observing performance changes after eliminating specific quantities of easy samples in a particular setting, which lacks scientific rigor. A more robust conclusion would require comprehensive theoretical proof or a broader range of experiments. A plausible hypothesis might be that easy samples contribute minimal information to the original dataset, and their outright removal could lead to a loss of information. A more balanced approach would be to prioritize sampling of 'hard' samples, as they contain richer information. This is the strategy employed by Dream, which explains its superior performance.

6. The setting of the parameter $\epsilon$ appears arbitrary and lacks meaningful justification.  The parameter study of $\epsilon$ shown in Figures verifies that 1.0 is the optimal value. However, this is simply the default setting used in the original MixUp method, suggesting a lack of innovation or significant analysis in choosing this particular value. $\epsilon$ will be more meaninful if there exists a setting that the optimal value is not 1.0, showing the difference with the default value used in MixUp.

**Questions:**

The questions are listed as the weaknesses in the previous section.

[1] Liu, Yanqing, et al. "DREAM: Efficient Dataset Distillation by Representative Matching." arXiv preprint arXiv:2302.14416 (2023).

[2] Liu, Songhua, et al. "Dataset distillation via factorization." Advances in Neural Information Processing Systems 35 (2022): 1100-1113.

[3] Yin, Zeyuan, Eric Xing, and Zhiqiang Shen. "Squeeze, Recover and Relabel: Dataset Condensation at ImageNet Scale From A New Perspective." arXiv preprint arXiv:2306.13092 (2023).

[4] Rui Song, Dai Liu, Dave Zhenyu Chen, Andreas Festag, Carsten Trinitis, Martin Schulz, and Alois
Knoll. Federated learning via decentralized dataset distillation in resource-constrained edge
environments. arXiv preprint arXiv:2208.11311, 2022.

---

### Author Response · Authors · 2023-11-16
**Update Manuscript**

We thank all the reviewers for helping us improve the paper. We uploaded a revised version of our manuscript and marked the major changes in blue. In short,

1. We have included more discussion of why dropping easy samples can help in Appendix B.

2. We have added more comparisons and discussion with more studies in related work and Appendix G.

Thank you all again for your valuable and insightful suggestions. Please let us know if you have additional questions or ideas for improvement.

Kind regards,
Authors

---

> ### Author Response · Authors · 2023-11-21
> **New updates in revision**
>
> We thank all the reviewers for helping us improve the paper. We just uploaded a newly revised version of our manuscript and marked the major changes in blue. In short,
>
> 1. We have included more discussion of why dropping easy samples can help in Appendix B. We have also included an example showing the differences between choosing the center vs focusing on harder samples in Figure 9. Many thanks to reviewer dBFm for helping us improve our paper.
> 2. We have added more comparisons and discussion with more studies in related work and Appendix G.
>
> As the deadline of discussion period is getting closer, we hope other reviewers can reply to us whether the concerns have been addressed appropriately. We are more than willing to discuss with you.
>
> Kind regards,
> Authors

---

### Meta-Review · Area_Chair_YhWd · 2023-12-05

**Metareview:**

1.	The definition of 'hard' or ‘easy’ examples in submission depends on their loss (line 6 in Algorithm 1), a concept that shares large similarity with the error vector score presented in [1], specifically,
Error vector score in [1]:    $\|\|p(\theta_t, \bf{x}) -\bf{y})\|\|_2$  and Loss score in submission:   $CE(p(\theta_t, \bf{x}),\bf{y})$.
Both of them think easy samples (samples with smaller loss or error) can be removed. However, as stated by Reviewer wdbv, the paper's conclusion that discarding easy samples is beneficial is based merely on observing performance changes after eliminating specific quantities of easy samples in a particular setting, which lacks scientific rigor. A more robust conclusion would require comprehensive theoretical proof or a broader range of experiments.

2.	As the paper introduces a sampling strategy to enhance dataset distillation, the comparison with the important baseline, Dream, appears to be insufficient. The limited experimental results presented in Table 9 indicate that the accuracy improvement of IDC+ISA over DEARM [3] is marginal or even inferior. Specifically, for CIFAR-10, the IDC+ISA method shows a 1% lower accuracy than DEARM [3] (73.8% compared to 74.8% for DREAM) with 50 IPC. Besides, there seems to be an unfair experimental setting when compared with Dream. The authors conduct ISA in an online manner, while Dream is executed offline (although the original paper of Dream claims it can be implemented online). It appears that the authors may be deliberately avoiding this to favour their own experimental results.

[1] Deep Learning on a Data Diet: Finding Important Examples Early in Training. NeurIPS 2021.

[2] Coverage-Centric Coreset Selection For High Pruning Rates. ICLR 2023.

[3] DREAM: Efficient Dataset Distillation by Representative Matching. ICCV2023.

**Justification For Why Not Higher Score:**

See above

**Justification For Why Not Lower Score:**

NA

---

### Decision · Program_Chairs · 2024-01-16

Reject